# The neuropeptide F/nitric oxide pathway is essential for shaping locomotor plasticity underlying locust phase transition

Li Hou[1], Pengcheng Yang[2], Feng Jiang[2], Qing Liu[1], Xianhui Wang[1]*, Le Kang[1,2]*

[1]State Key Laboratory of Integrated Management of Pest Insects and Rodents, Institute of Zoology, Chinese Academy of Sciences, Beijing, China; [2]Beijing Institutes of Life Science, Chinese Academy of Sciences, Beijing, China

**Abstract** Behavioral plasticity is widespread in swarming animals, but little is known about its underlying neural and molecular mechanisms. Here, we report that a neuropeptide F (NPF)/nitric oxide (NO) pathway plays a critical role in the locomotor plasticity of swarming migratory locusts. The transcripts encoding two related neuropeptides, NPF1a and NPF2, show reduced levels during crowding, and the transcript levels of NPF1a and NPF2 receptors significantly increase during locust isolation. Both NPF1a and NPF2 have suppressive effects on phase-related locomotor activity. A key downstream mediator for both NPFs is nitric oxide synthase (NOS), which regulates phase-related locomotor activity by controlling NO synthesis in the locust brain. Mechanistically, NPF1a and NPF2 modify NOS activity by separately suppressing its phosphorylation and by lowering its transcript level, effects that are mediated by their respective receptors. Our results uncover a hierarchical neurochemical mechanism underlying behavioral plasticity in the swarming locust and provide insights into the NPF/NO axis.

*For correspondence: wangxh@ioz.ac.cn (XW); lkang@ioz.ac.cn (LK)

**Competing interests:** The authors declare that no competing interests exist.

## Introduction

Swarming occurs in a wide variety of animal taxa, including insects, fish, birds, and mammals. Individuals benefit from swarming in many aspects, including food searching, territory selection, and defense (*Okubo, 1986*; *Weaver et al., 1989*). Typically, to maintain the required fission–fusion dynamics, swarming animals exhibit striking behavioral plasticity of different types (*Snell-Rood, 2006*; *Szyf, 2010*). Biochemical changes in the levels of neuromodulators, such as monoamines, neuropeptides, and neurohormones, are able to induce behavioral variation thus mediate behavioral plasticity (*Freudenberg et al., 2015*; *Godwin et al., 2015*; *Zupanc and Lamprecht, 2000*). Nevertheless, the molecular basis by which neural factors orchestrate behavioral plasticity in swarming animals is poorly understood in detail.

Neuropeptides, a group of chemically diverse neural modulators, affect a broad range of physiological and behavioral activities (*Lieberwirth and Wang, 2014*; *Nässel, 2002*). Accumulating evidence shows that neuropeptides serve as conserved neuronal signals that modulate animal behaviors in social contexts (*Lieberwirth and Wang, 2014*; *Nilsen et al., 2011*). These peptides exert their actions by binding to specific membrane receptors, most of which are G-protein-coupled receptors (*Quartara and Maggi, 1997*). The binding initiates a second-message cascade unique for each receptor and results in a distinct molecular response (*Hökfelt et al., 2003*). It has been revealed that neuropeptides can induce plasticity in a series of behavioral processes, including sensory detection (*Shankar et al., 2015*), signal integration (*Grammatopoulos, 2012*), and behavioral

**eLife digest** Migratory locusts are widespread throughout the Eastern Hemisphere, especially in Asia, Australia and Africa. Although usually solitary insects, locusts can also form swarms made up of millions of individuals, which can devastate crops. Swarming can be studied on a smaller scale in the laboratory by forcing locusts to crowd together. This causes the locusts to enter a so-called gregarious state in which they are more active and sociable, which in turn promotes swarming. Isolating individual locusts has the opposite effect, causing the insects to enter a solitary state in which they are less active.

Chemicals in the locust brain called neuropeptides control phase transitions between solitary and gregarious behavior. Neuropeptides bind to specific proteins called receptors in the outer membranes of neurons and initiate unique signaling cascades inside cells. However, exactly how neuropeptides regulate the changes in locust behavior that affect swarming was not clear.

Hou et al. now reveal the role that two related neuropeptides, NPF1a and NPF2, play in this process. Crowding causes the levels of NPF1a and NPF2 in the locust brain to decrease, whereas isolating individual locusts causes the levels of two NPF receptors to increase. Both neuropeptides reduce levels of a molecule called nitric oxide in the brain. NPF1a does this by reducing the activity of the enzyme that produces nitric oxide, whereas NPF2 reduces the production of this enzyme. The reduction in nitric oxide in turn makes the locusts less active.

Similar NPF neuropeptides had previously been shown to affect activity levels in other invertebrates, such as roundworms and fruit flies. This, combined with the results now presented by Hou et al., suggests that the NPF/nitric oxide pathway may regulate activity in insects in general. Future work should investigate this possibility, as well as whether the NPF/nitric oxide pathway controls changes in other insect behaviors such as feeding and mating.

responsiveness (*Ruzza et al., 2015*) by acting either individually or in concert with other neuromodulators (*Dölen et al., 2011*; *Flores et al., 2015*; *Maroun and Wagner, 2016*). Therefore, neuropeptides and their downstream components may act as vital parts of the regulatory network underlying behavioral plasticity in swarming animals.

The migratory locust, *Locusta migratoria*, exhibits two interconvertible phases, the solitarious phase (S-phase) and the gregarious phase (G-phase), the latter of which is characterized by swarming behavior (*Ariel and Ayali, 2015*). Locust behaviors in the two phases significantly differ, most notably in the interaction among individuals and in locomotor activity (*Uvarov, 1977*). S-phase locusts are sedentary and repel their conspecifics, whereas G-phase individuals are highly active and attract their conspecifics (*Simpson et al., 1999*). The behavioral transition between two phases is promoted by either isolating G-phase locusts (that is, solitarization) or, in the opposite direction, by forced crowding of S-phase locusts (that is, gregarization), the key step in seeding locust swarming (*Pener and Simpson, 2009*). Behavioral solitarization occurs faster than behavioral gregarization in the migratory locust. The attraction index and locomotor activity of locusts continuously decrease within 16 hr after isolation. By contrast, these behaviors do not increase until 32 hr after crowding, but are far below the level of gregarious controls even after crowding for 64 hr (*Guo et al., 2011*).

The locust brain undergoes strong neurochemical reconfiguration during behavioral phase transition; for instance, the contents of several neurotransmitters that mediate synaptic plasticity show significant change (*Rogers et al., 2004*; *Ma et al., 2011*, *2015*). Recently, we have found that several neuropeptide genes are differentially expressed between the central nervous systems of G-phase and S-phase locusts (*Hou et al., 2015*), suggesting possible modulatory roles for these neuropeptides in the behavioral phase transition.

Here, we show that two related neuropeptides, NPF1a and NPF2, act as crucial neural modulators in the phase-related locomotor plasticity of the migratory locust. We uncover a potentially important connection between the atypical neurotransmitter NO and the two NPFs, a connection mediated by NOS. We therefore suggest that the actions of NPFs (or their homolog NPY) may be mediated, partly through NOS and NO, in other organisms.

# Results

## Two related neuropeptides, NPF1a and NPF2, affect phase-related locomotor activity

We have previously shown that 15 neuropeptide-encoding genes are differentially expressed in the brains of G-phase and S-phase locusts (*Hou et al., 2015*). Here, we extend our work to explore which of these neuropeptides are closely tied to the behavioral phase transition. qPCR analysis (*Figure 1* and *Figure 1—figure supplement 1*) revealed that the mRNA levels of four neuropeptide encoding genes, namely, *AKH/Corazonin related peptide* (*ACP*), *Insulin-like peptide* (*ILP*), *NPF1a*, and *NPF2*, significantly changed in the phase transition, that is, during solitarization or gregarization or both. During gregarization, the mRNA levels of *ACP* and *ILP* steadily increased, whereas those of *NPF1a* and *NPF2* rapidly decreased. During solitarization, the transcript levels of *ILP* and *NPF1a* significantly changed compared to those of *ACP* and *NPF2*.

To assess whether these four neuropeptides are involved in the behavioral phase transition, we performed a behavioral screen in the G-phase locusts using transcript knockdown or peptide

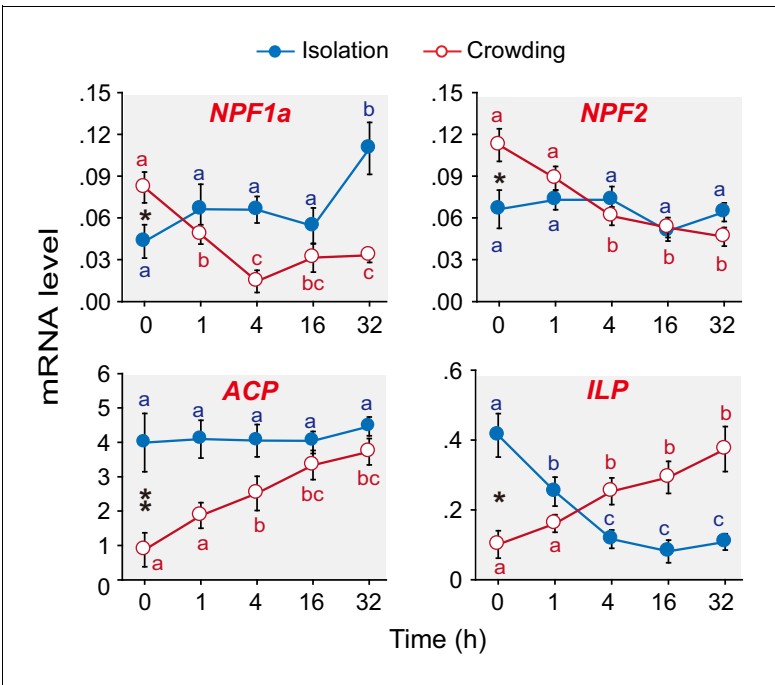

**Figure 1.** Levels of transcripts encoding the neuropeptides NPF1a, NPF2, ACP and ILP change during the G/S phase transition in the migratory locust. qPCR was performed to determine the transcript levels of 15 neuropeptide-encoding genes in locust brains in the time course of the isolation of gregarious (G-phase) locusts or the crowding of solitarious (S-phase) locusts. Four neuropeptide genes displayed clear expression changes during isolation or crowding or both (in the case of *NPF1a* and *ILP*). Raw data measuring the mRNA levels of the four neuropeptide genes are shown in *Figure 1—source data 1*. For the transcript levels of the other 11 neuropeptide genes, see *Figure 1—figure supplement 1*. The data are presented as mean ± s.e.m. Significant differences at different times are denoted by letters (n = 4 samples per timepoint, 8 animals/sample, one-way ANOVA, p<0.05). *indicates a significant difference between typical G-phase (0 hr after isolation) and typical S-phase (0 hr after crowding) locust brains (Student's *t*-test, *p<0.05, **p<0.01).

The following source data and figure supplement are available for figure 1:

**Source data 1.** mRNA levels of the four neuropeptide-encoding genes during isolation and crowding processes.

**Figure supplement 1.** The transcript levels of 11 neuropeptides do not change during the G/S phase transition in the migratory locust.

injection. The behavioral phase state was then assessed in an arena assay and measured by $P_{greg}$, which is calculated using a binary logistic regression model that retains three variables: attraction index, total distance moved, and total duration of movement (*Guo et al., 2011*). $P_{greg}$ varies between 0 (in the fully S-phase behavioral state) and 1 (in the fully G-phase behavioral state). We performed RNAi-mediated transcript knockdown to reduce the levels of *ACP* and *ILP*, which show higher transcript levels in G-phase locust brains (*Figure 1*, lower). We found that knockdown of either *ACP* or *ILP* transcript did not significantly change the $P_{greg}$ values of G-phase locusts (*Figure 2—figure supplement 1*). On the other hand, we injected synthetic peptides to increase the concentrations of NPF1a and NPF2, which display lower transcript levels in G-phase locust brains (*Figure 1*, upper). G-phase locusts that were injected with NPF1a or NPF2 peptide behaved in a way that became considerably more solitarious, in a dose-dependent manner, when compared to control locusts (*Figure 2A* and *Figure 2—figure supplement 2A*). Co-injection of both NPF1a and NPF2 peptides into G-phase locusts enhanced the reduction of $P_{greg}$ compared to that seen following the injection of either NPF peptide alone (*Figure 2A*). Moreover, injection of NPF1a peptide provoked a faster inhibitory effect on the $P_{greg}$ values of locusts than that caused by NPF2 peptide injection (*Figure 2B* and *Figure 2—figure supplement 2B*). However, G-phase locusts that were injected with either ds*NPF1a* or ds*NPF2* or with a mixture of these constructs did not show significant behavioral changes relative to control locusts (*Figure 2—figure supplement 3*, left).

We validated the roles of two NPFs in the behavioral change in S-phase locusts by transcript knockdown of *NPF1a* and *NPF2* individually or together. S-phase locusts that were injected with ds*NPF1a* displayed a significant behavioral change in the direction of G-phase, whereas injection of ds*NPF2* did not significantly change the $P_{greg}$ values (*Figure 2C* and *Figure 2—figure supplement 4*). However, the S-phase locusts that were injected with either ds*NPF1a* or ds*NPF2* were more gregarious than the controls in response to crowding stimuli, and these effects were strengthened by the dual-knockdown of the *NPF1a* and *NPF2* transcripts (*Figure 2D*). Furthermore, peptide injection of NPF1a or NPF2 or their mixture in S-phase locusts did not affect their behavioral phase states (*Figure 2—figure supplement 3*, right).

Behavioral parameter analysis demonstrated that locust locomotor activity, including total duration of movement and total distance moved, were strongly suppressed by the treatments that increased the levels of NPF1a or NPF2 peptide in G-phase locusts, but enhanced by ds*NPF1a* or ds*NPF2* injection in S-phase locusts (*Figure 2E–H*), while the attraction index was not significantly changed by these treatments (*Figure 2—figure supplement 5*). Thus, NPF1a and NPF2 play important roles in the locust behavioral phase transition by modulating locomotor activity.

## Two NPF receptors, NPFR and NPYR, are essential for changes in locomotor activity related to the phase transition

Bioinformatically, we obtained two locust sequences with high similarity to the *Drosophila NPFR* gene (*Supplementary file 1*). They were named *LomNPFR* and *LomNPYR*, based on their phylogenetic relationship with homologs in other species (*Figure 3—figure supplement 1B*). Competitive binding experiments indicated that NPF1a peptide displayed much higher affinity to HEK 293 T cells expressing NPFR protein ($IC_{50}$ = 24 nM) than did NPF2 peptide ($IC_{50}$ = 355 nM) (*Figure 3A* and *Figure 3—figure supplement 2*), whereas NPF2 displayed much higher affinity to NPYR-expressing cells ($IC_{50}$ = 64.5 nM) than did NPF1a ($IC_{50}$ = 380 nM) (*Figure 3B*).

The mRNA level of *NPFR* increased greatly within 1 hr after isolation of G-phase locusts, whereas it showed no change during locust crowding (*Figure 3C*). By contrast, the transcript level of *NPYR* responded to both isolation and crowding, with an obvious increase during isolation and a significant reduction during crowding (*Figure 3D*). Transcript knockdown of either *NPFR* or *NPYR* facilitated the transition from S-phase traits towards G-phase traits by influencing the locomotor activity of locusts (*Figure 3E,F* and *Figure 3—figure supplement 3*). Moreover, the dual-knockdown of *NPFR* and *NPYR* significantly strengthened the enhancement of both total distance moved and total duration of movement caused by knockdown of either transcript individually (*Figure 3E,F*). These results suggest that these two NPF receptors are essential for the regulation of phase-related locomotor activity.

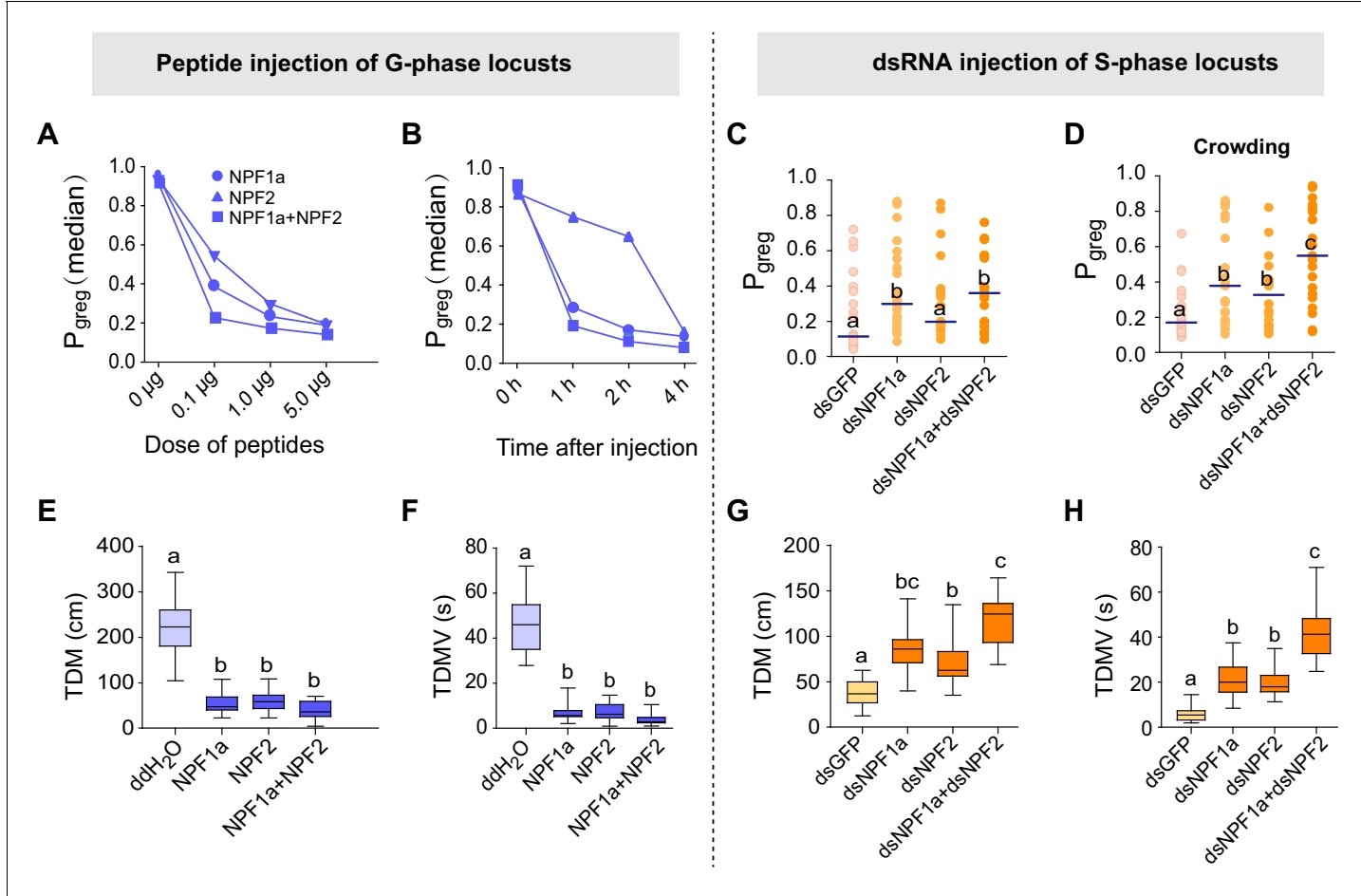

**Figure 2.** Perturbations of NPF1a or NPF2 peptide levels or of their transcript levels leads to changes in locomotor activity related to the G/S phase transition. Locust behaviors are measured by the term $P_{greg}$, which is a combined assessment of movement and inter-insect attraction (indicated as attraction index, see *Figure 2—figure supplement 2*). $P_{greg} = 0$ represents a fully S-phase behavioral state; $P_{greg} = 1$ represents a fully G-phase behavioral state. (A) and (B) Dose- and time-dependent changes in the median $P_{greg}$ of G-phase locusts after injection of NPF1a and NPF2 peptides, separately and together. For detailed $P_{greg}$ distributions and statistics, see *Figure 2—figure supplement 2* (n ≥ 18 locusts, Mann–Whitney U test, p<0.05). (C) $P_{greg}$ in S-phase locusts 48 hr after transcript knockdown of *NPF1a*, or *NPF2*, or both (n ≥ 20 locusts, Mann–Whitney U test, p=0.020, 0.064 and 0.017, respectively). Lines indicate median $P_{greg}$. Significant differences are denoted by letters. (D) $P_{greg}$ in crowded S-phase locusts after transcript knockdown of *NPF1a*, or *NPF2*, or both (n ≥ 20 locusts, Mann–Whitney U test, p=0.024, 0.039 and 0.037, respectively). Locusts were forced into a crowd 32 hr after dsRNA injection, and their behaviors were measured after 16 hr of crowding (that is 48 hr after dsRNA injection). (E) and (F) Total distance moved (TDM) and total duration of movement (TDMV) 4 hr after injection of NPF1a or NPF2 or both peptides in G-phase locusts (5 µg/ individual). The data are presented as mean ± s.e.m. Significant differences are denoted by letters (n ≥ 18 locusts, one-way ANOVA, p<0.05). (G) and (H) Total distance moved (TDM) and total duration of movement (TDMV) 48 hr after transcript knockdown of *NPF1a* or *NPF2* or both genes in S-phase locusts (n ≥ 20 locusts).

The following figure supplements are available for figure 2:

**Figure supplement 1.** Transcript knockdown of *ACP* or *ILP* does not significantly affect behavioral phase state in G-phase locusts.

**Figure supplement 2.** Injection of NPF1a or NPF2 peptide into G-phase locusts induces S-phase-like behaviors in a dose- and time-dependent manner.

**Figure supplement 3.** Transcript knockdown of *NPF1a* or *NPF2* in G-phase locusts and peptide injection of NPF1a or NPF2 in S-phase locusts do not affect phase-related behaviors.

**Figure supplement 4.** Efficiency and specificity of *NPF1a* and *NPF2* transcript knockdown.

*Figure 2 continued on next page*

*Figure 2 continued*

**Figure supplement 5.** Perturbation of NPF1a or NPF2 peptide, or of their transcript levels, do not change attraction index related to the G/S phase transition.

## NO signaling is a downstream component under the regulation of NPF1a and NPF2

To explore how NPF1a and NPF2 regulate locomotor plasticity during the G/S phase transition, we analyzed RNAseq-based transcriptomic differences in three comparisons: G-phase and S-phase locusts (comparison 1: C1); co-injection of NPF1a and NPF2 peptides in G-phase locusts with control injection (comparison 2: C2); co-injection of ds*NPF1a* and ds*NPF2* in S-phase locusts with control injection (comparison 3: C3). We identified a total of 221, 317, and 313 differentially expressed genes in the three comparisons, respectively (*Figure 4—figure supplement 1A*), and 32% of these genes were annotated (*Figure 4—figure supplement 1B*). Numerous differentially expressed genes encoding catalytic and binding activities were clearly enriched in each treatment (*Figure 4—figure supplement 1C*).

A number of genes displayed altered transcription patterns (*Figure 4A*) that are consistent with locust behavioral change caused by the manipulation of NPF1a and NPF2 levels, as shown in *Figure 2*. The transcript levels of these genes were different between the typical G-phase and S-phase locusts (C1). Moreover, their transcript levels changed oppositely in the two treatments: co-injection of NPF1a and NPF2 peptides in G-phase locusts (C2) and dual-knockdown of *NPF1a* and *NPF2* transcripts in S-phase locusts (C3). Among these genes, we found that several genes encode important signaling molecules. Using qPCR, the expression patterns of two genes, adenylate cyclase (*AC2*) and *NOS*, were confirmed in all three comparisons (*Figure 4B* and *Figure 4—figure supplement 1D*). The two genes showed high transcript levels in the brains of G-phase locusts. Moreover, their transcript levels were significantly lower after the co-injection of NPF1a and NPF2 peptides in G-phase locusts, and were increased by dual-knockdown of *NPF1a* and *NPF2* transcripts in S-phase locusts (*Figure 4B* and *Figure 4—figure supplement 1D*).

AC2 catalyzes cAMP production and might activate the PKA pathway, whereas NOS catalyzes NO production resulting in the activation of NO signaling (*Mete and Connolly, 2003*; *Watts and Neve, 1997*). We therefore examined whether cAMP and NO levels could be influenced by the manipulation of NPF1a and NPF2 levels. NO concentration in brains decreased dramatically within 4 hr after injection of NPF1a or NPF2 or of the peptide mixture into G-phase locusts, and significantly increased after knockdown of *NPF1a* or *NPF2* or both *NPF* transcripts in S-phase locusts (*Figure 4C*). By contrast, there was no change in cAMP level 4 hr after manipulation of either NPF1a or NPF2 level (*Figure 4—figure supplement 2*). These data suggest that NO signaling may serve as a downstream pathway for both NPFs in the locust.

## NO signaling acts as vital stimulator of locomotor activity in the G/S phase transition

The mRNA and protein levels of NOS were considerably higher in G-phase than in S-phase locust brains (*Figure 5A,B*), and significantly changed during the G/S phase transition (*Figure 5C–E*). Interestingly, NOS was present in both phosphorylated and non-phosphorylated forms (*Figure 5—figure supplement 1A–C*). Phosphorylated NOS was more abundant in the brains of G-phase locusts than in those of S-phase locusts (*Figure 5B*). Dephosphorylation of NOS by λ-phosphatase significantly reduced NOS activity and the NO level (*Figure 5—figure supplement 1D,E*). During the G/S phase transition, the level of NOS phosphorylation decreased or increased within 4 hr after solitarization or gregarization, respectively (*Figure 5D,E*). These changes occurred much faster than the alterations in *NOS* mRNA level, which did not change until 16 hr after solitarization or gregarization (*Figure 5C*). In addition, NO levels in the locust brains continuously decreased during solitarization, but sharply increased 32 hr after gregarization (*Figure 5F*). The changes in NO levels are tightly linked to the G/S behavioral phase transition.

We then conducted a series of molecular, pharmacological and behavioral experiments to investigate the function of NO signaling in the G/S locust phase transition. Knockdown of the *NOS*

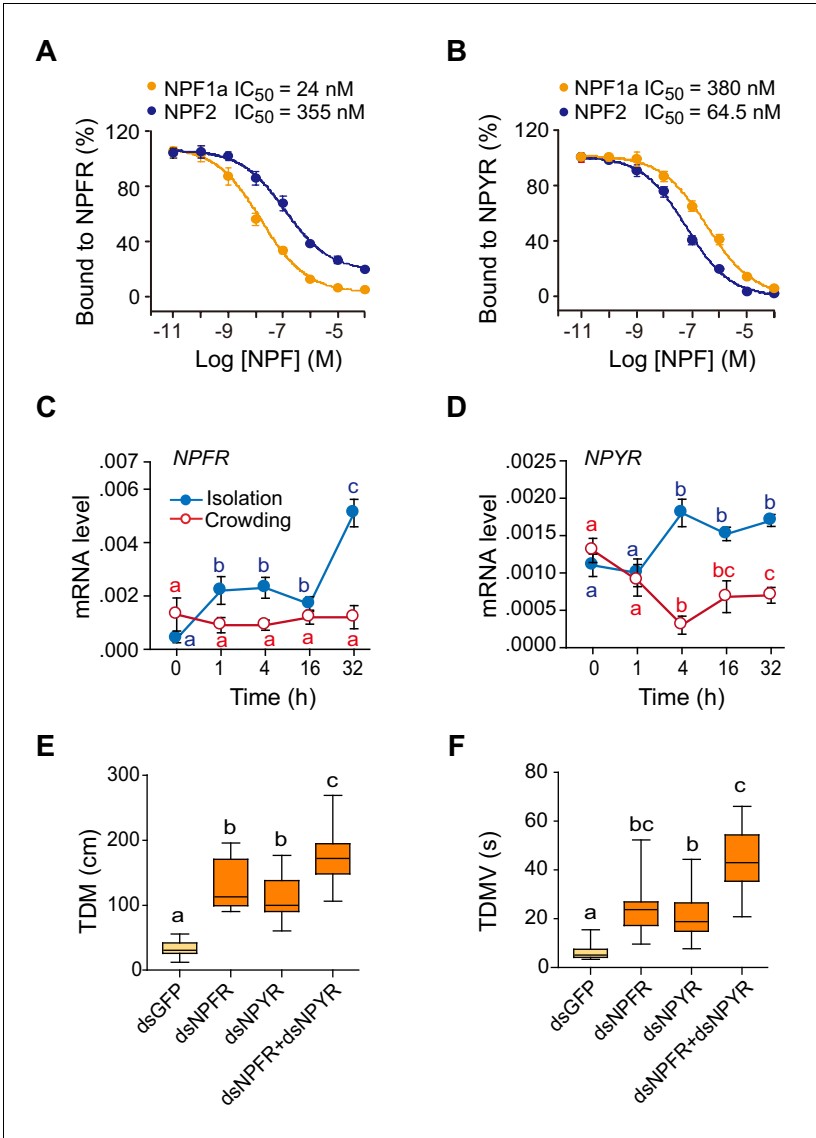

**Figure 3.** Receptors for NPF1a and NPF2 are involved in transmitting the effects of these neuropeptides on locomotor activity. (**A**) Competitive inhibition of TAMRA-NPF1a binding to HEK 293 T cells transfected with pcDNA3.1-NPFR vector (n = 6). (**B**) Competitive inhibition of TAMRA-NPF2 binding to HEK 293 T cells transfected with pcDNA3.1-NPYR vector (n = 6). (**C**) and (**D**) Time course patterns of *NPFR* and *NPYR* transcript levels during the G/S locust phase transition (isolation, shown in blue; crowding, shown in red). The data are presented as mean ± s.e.m (n = 4 samples per timepoint, 8 locusts/sample, one-way ANOVA, $p < 0.05$). Detailed expression levels of the two NPF receptors are shown in *Figure 3—source data 1*. (**E**) and (**F**) Total distance moved (TDM) and total duration of movement (TDMV) 48 hr after transcript knockdown of *NPFR* or *NPYR* or both genes in S-phase locusts. Significant differences are denoted by letters (n ≥ 19 locusts, one-way ANOVA, $p < 0.05$).

The following source data and figure supplements are available for figure 3:

**Source data 1.** Transcript levels of *NPFR* and *NPYR* during the G/S locust phase transition.

**Figure supplement 1.** Phylogenetic relationship of NPF or NPY precursors and their receptors in different species.

**Figure supplement 2.** Overexpressions of (**A**) NPFR and (**B**) NPYR in HEK 293 T cells validated by western blot.

**Figure supplement 3.** Transcript knockdown of *NPFR* or *NPYR* in S-phase locusts induces G-phase-like behaviors without affecting attraction index.

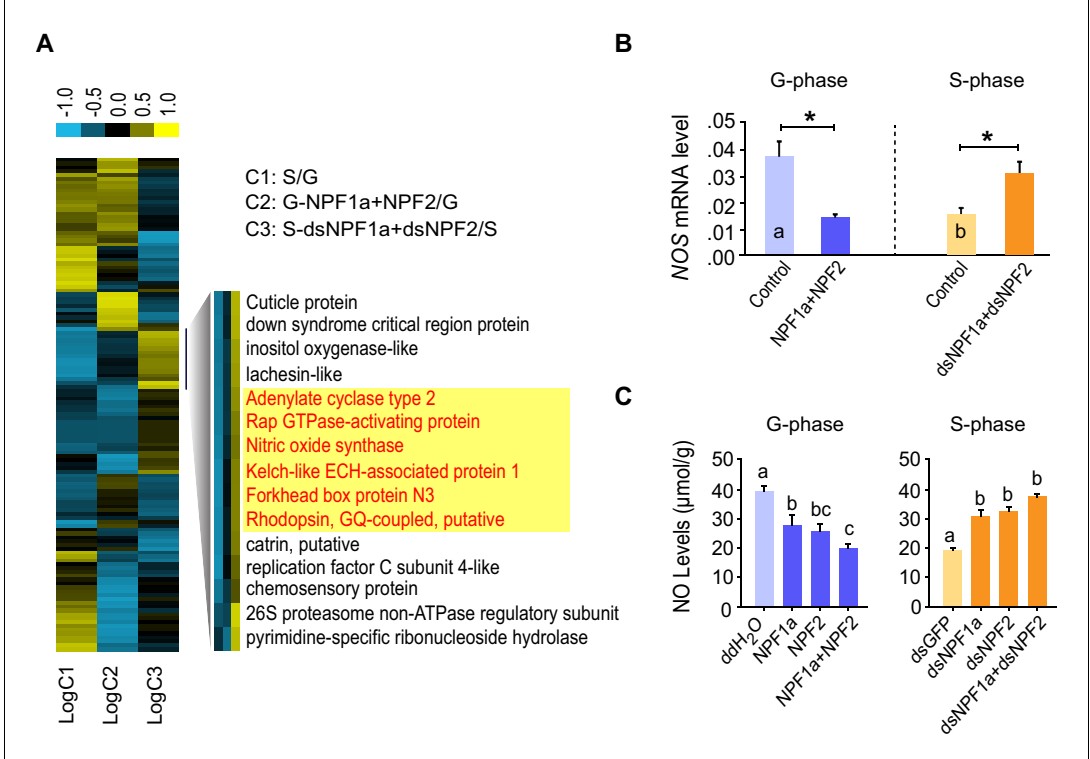

**Figure 4.** Cluster analysis of RNA-Seq data leads to the identification of nitric oxide synthase (NOS) as a downstream component of the NPF1a and NPF2 pathway. (A) Cluster analysis of differentially expressed genes in the transcriptome. Several important genes (highlighted in yellow) involved in signaling in other organisms display expression patterns that correlate with behavioral change after the manipulation of NPF1a and NPF2 peptides or transcript levels. Logarithmic fold alteration of treatment versus control is shown in the heat map. Yellow and blue colors indicate up- and downregulation, respectively (n = 3 samples per treatment, 10 animals/sample). For detailed gene-expression data, please see *Figure 4—source data 1*. (B) Transcript levels of *NOS* in the brains after co-injection of NPF1a and NPF2 peptides in G-phase locusts or transcript knockdown of both *NPF1a* and *NPF2* in S-phase locusts (n = 5 samples, 8 locusts/sample, Student's *t*-test, *p<0.05, different letters labeled in columns indicate a significant difference). (C) NO levels after injection of NPF1a and NPF2 peptides, separately and together, in G-phase locusts, or after transcript knockdown of *NPF1a* and *NPF2*, separately and together, in S-phase locusts. The data are presented as mean ± s.e.m. Significant differences are denoted by letters (n = 4 samples, 16 locusts/sample, one-way ANOVA, p<0.05).

The following source data and figure supplements are available for figure 4:

**Source data 1.** The effects of NPF1a and NPF2 on the expression of annotated genes in the brains of fourth-instar locusts.

**Figure supplement 1.** Transcriptomic profiles influenced by NPF1a and NPF2 in locust brains revealed by RNA-seq.

**Figure supplement 2.** cAMP levels after artificial manipulation of NPF1a or NPF2 peptide or their transcript levels.

transcript or injection of the NOS inhibitor N-Nitro-L-arginine Methyl Ester (L-NAME) into G-phase locusts strongly suppressed the total duration of movement and total distance moved (*Figure 6A–D*), thus resulting in S-phase-like behavior (*Figure 6—figure supplement 1A,C*). By contrast, injection of S-phase locusts with the NO donor S-nitroso-N-acetyl-penicillamine (SNAP) enhanced the total duration of movement and total distance moved (*Figure 6E,F*), and pushed locust behavioral change from S-phase to G-phase state (*Figure 6—figure supplement 1E*). All manipulations (including *NOS* transcript knockdown and injections of the two chemicals) did not change the attraction index of tested locusts (*Figure 6—figure supplement 1B,D,F*). Furthermore, both transcript knockdown and L-NAME injection significantly reduced NOS activity and NO levels in G-phase locust brains, whereas SNAP injection increased NO levels in S-phase locust brains without affecting NOS activity (*Figure 6—figure supplement 2*).

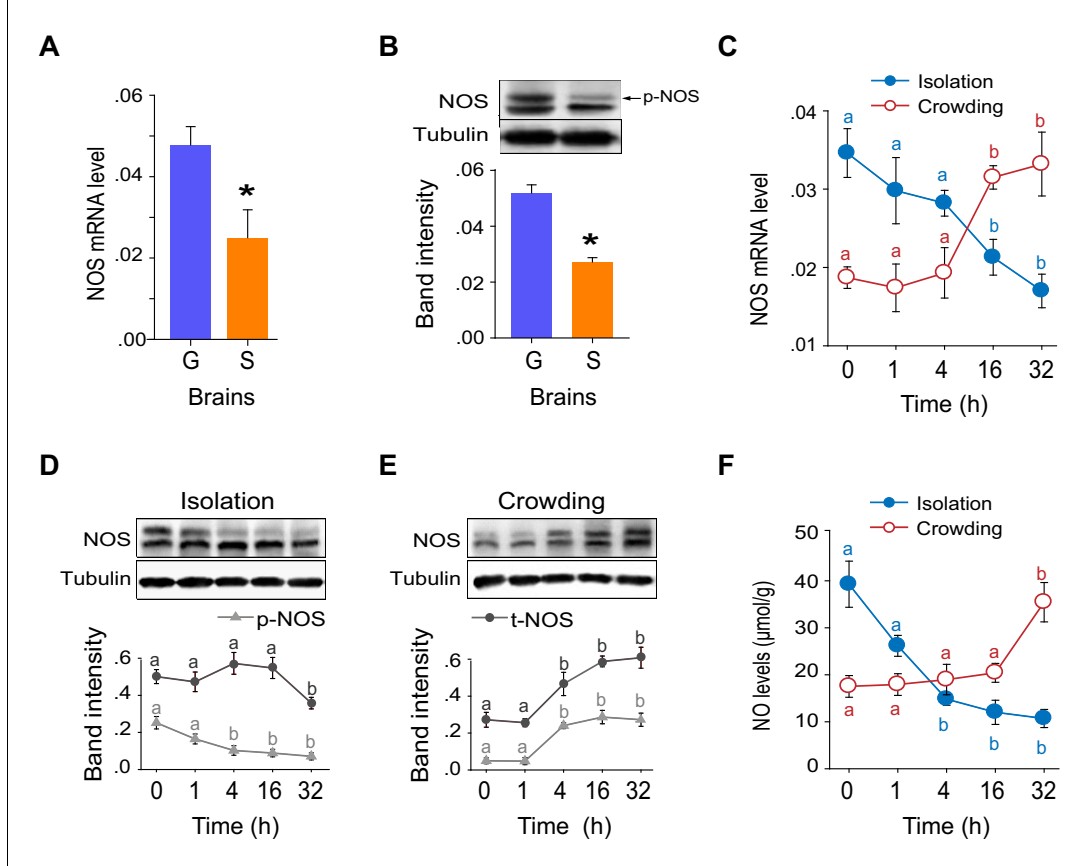

**Figure 5.** NOS transcript levels and phosphorylation states and NO levels differ in G-phase and S-phase locust brains. (A) *NOS* mRNA levels in the brains of G-phase and S-phase locusts (n = 4 samples, 8 locusts/sample, Student's *t*-test, *p<0.05). (B) NOS protein levels in the brains of G-phase and S-phase locusts. The upper band detected by anti-uNOS indicates phosphorylated NOS (p-NOS, see *Figure 5—figure supplement 1*) (n = 3 samples, 12 locusts/sample, Student's *t*-test, *p<0.05). (C) Time course of *NOS* mRNA levels during the G/S phase transition (n = 4 samples/timepoint, 8 locusts/sample, one-way ANOVA, p<0.05, isolation shown in blue; crowding shown in red). (D) and (E) Time course of NOS protein levels during the G/S phase transition (n = 3 samples, 10–12 locusts/sample, phosphorylated NOS data are represented by triangles; total NOS data are represented by dots). The protein level is referenced to β-tubulin. (F) Time course of NO levels during the G/S phase transition. All data are presented as mean ± s.e.m. Significant differences are denoted by letters (n = 4 samples, 16 locusts/sample, one-way ANOVA, p<0.05). Raw data showing the changes in NOS mRNA level, NOS protein level and NO level are shown in *Figure 5—source data 1*.

The following source data and figure supplement are available for figure 5:

**Source data 1.** Time-course changes in NOS mRNA level, NOS protein level and NO level during the G/S phase transition.

**Figure supplement 1.** Reducing NOS expression and reducing NOS phosphorylation levels decrease NOS activity and NO level.

## NPF1a and NPF2 sequentially suppress NO signaling at the phosphorylation and transcription levels

We have shown that NO levels were decreased by injection of either NPF1a or NPF2 and increased by knockdown of *NPF1a* or *NPF2* transcripts (*Figure 4C*). Next we asked whether the two NPFs suppress the NO signaling pathway. The mRNA and protein levels of NOS significantly decreased 4 hr after injection of NPF2 peptide into G-phase locusts (*Figure 7B,E*). On the other hand, the mRNA and protein levels of NOS increased after knockdown of the *NPF2* transcript in S-phase locusts (*Figure 7C,F*). By contrast, no change in *NOS* mRNA level was observed in any treatments involving NPF1a (*Figure 7A,C*). However, the level of phosphorylated NOS significantly decreased 1 hr after injection of NPF1a peptide into G-phase locusts (*Figure 7D*) and increased after knockdown of the *NPF1a* transcript in S-phase locusts (*Figure 7F*). Injection of NPF1a or NPF2 peptide into G-phase

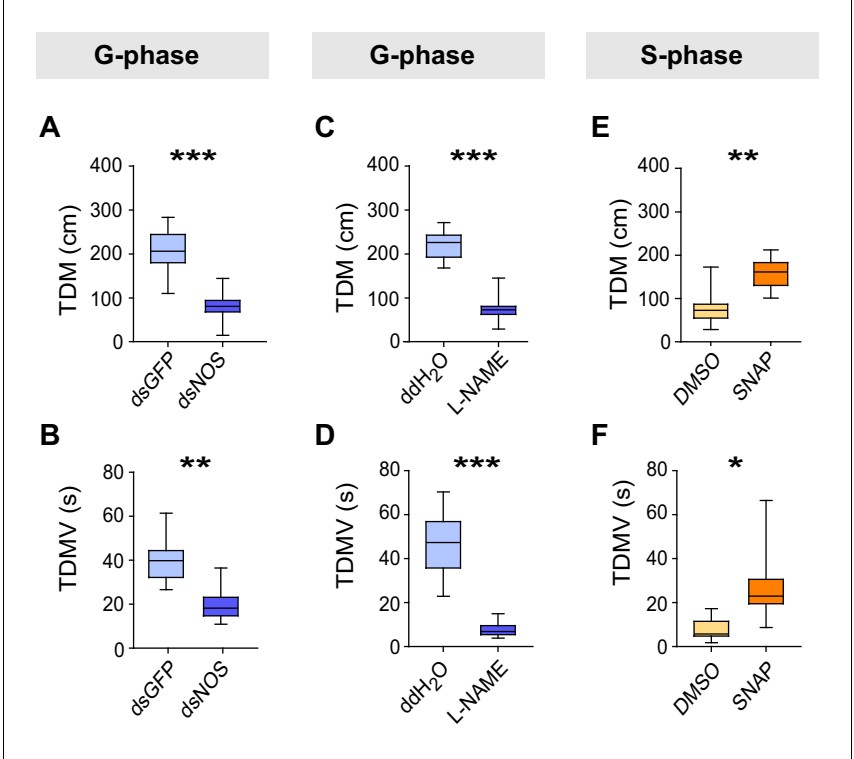

**Figure 6.** Perturbations of NO levels by transcript knockdown and drug injection dramatically change G-phase and S-phase locust behaviors. (A) and (B) Total distance moved (TDM) and total duration of movement (TDMV) of G-phase locusts 48 hr after knockdown of the *NOS* transcript. All data are presented as mean ± s.e.m. (n ≥ 23 locusts, Student's *t*-test, *p<0.05, **p<0.01, ***p<0.001). (C) and (D) Total distance moved (TDM) and total duration of movement (TDMV) of G-phase locusts 2 hr after injection of NOS inhibitor (L-NAME). (E) and (F) Total distance moved (TDM) and total duration of movement (TDMV) of S-phase locusts 2 hr after injection of NO donor (SNAP).

The following figure supplements are available for figure 6:

**Figure supplement 1.** Effects on $P_{greg}$ and attraction index after *NOS* transcript knockdown and drug treatments in G-phase and S-phase locusts.

**Figure supplement 2.** Effects on NOS activity and NO levels after *NOS* transcript knockdown and drug treatments in G-phase and S-phase locusts.

locusts significantly decreased NOS activity and NO levels in a time-dependent manner, with NPF1a exhibiting an earlier inhibitory effect on NO signaling than NPF2 (*Figure 7G,H*). Conversely, knockdown of *NPF1a* or *NPF2* enhanced NOS activity in S-phase locusts (*Figure 7I*), which is consistent with the changing patterns of NO levels in the same treatments (*Figure 4C*). These data further verify the effects of NPF1a and NPF2 on NOS/NO signaling.

## NPF1a and NPF2 co-localize with NOS in the pars intercerebralis

To understand the neural basis for the interactions between two NPFs and NO signaling in the regulation of phase-related locomotion, we localized NOS and the two NPF peptides in the locust brain by double immunofluorescence staining. NOS was extensively expressed in the cell bodies of neurons in the pars intercerebralis and in the Kenyon cells anterior to the calyces of mushroom bodies in each brain hemisphere (*Figure 8* and *Figure 8—figure supplement 1*). The distribution of NPF1a peptide was similar to that of NOS. NPF1a and NOS were co-localized in two regions, namely, the pars intercerebralis (*Figure 8*, upper) and the pars lateralis anterior to the calyces of mushroom bodies (*Figure 8—figure supplement 1*). However, NPF2 showed co-localization with NOS only in the

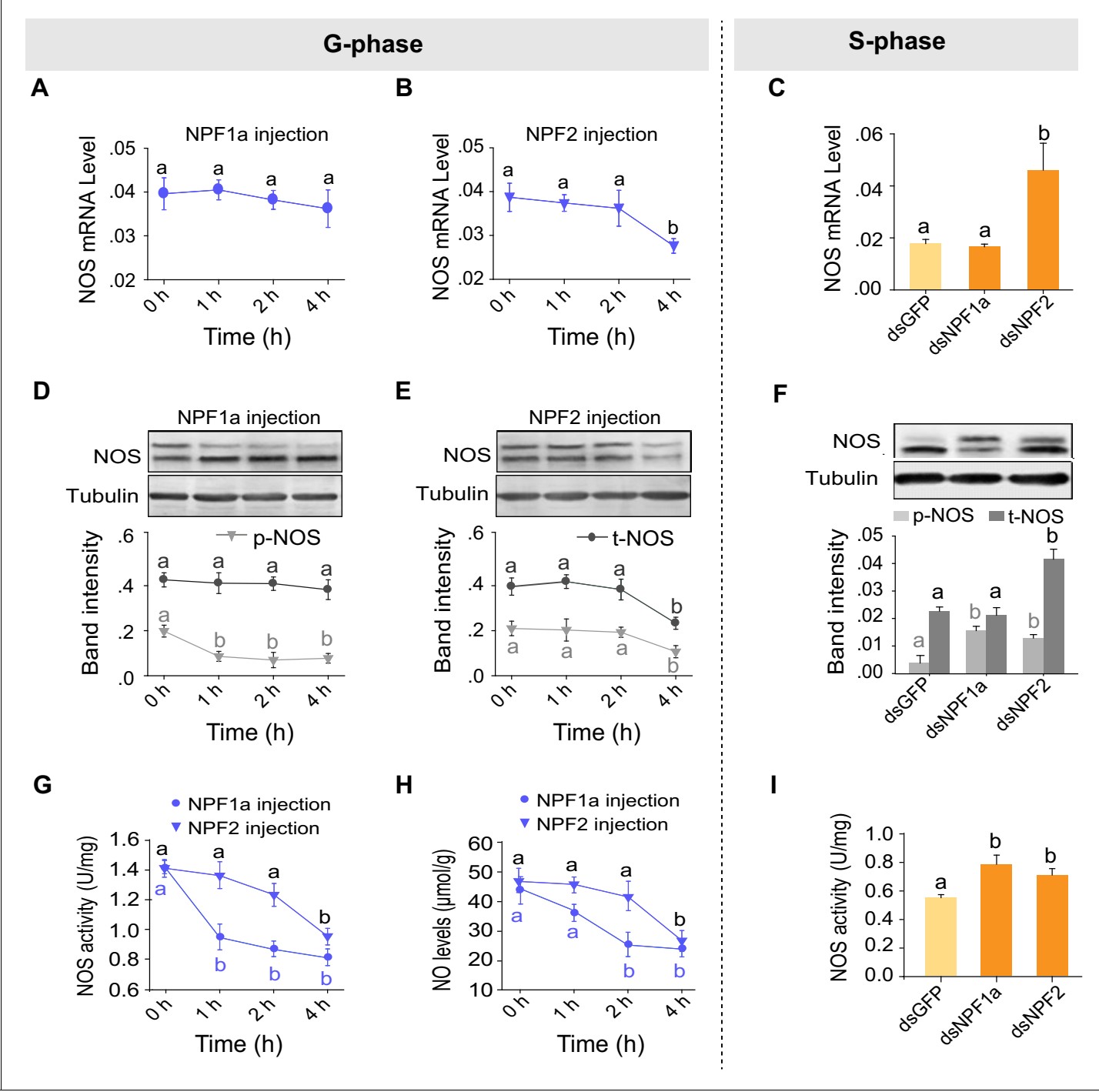

**Figure 7.** Manipulations of NPF1a and NPF2 levels alter NOS activity and phosphorylation states in the brains of G-phase and S-phase locusts. (A) and (B) *NOS* mRNA levels after injection of NPF1a or NPF2 peptide into G-phase locusts. The data are presented as mean ± s.e.m. Significant differences are denoted by letters (n = 4 samples, 8 locusts/sample, one-way ANOVA, p<0.05). (C) *NOS* mRNA levels 48 hr after transcript knockdown of *NPF1a* or *NPF2* in S-phase locusts (n = 4 samples, one-way ANOVA, p<0.05). (D) and (E) NOS protein levels after injection of NPF1a or NPF2 peptide into G-phase locusts (n = 3 samples, 10–12 locusts/sample, one-way ANOVA, p<0.05). (F) NOS protein levels 48 hr after transcript knockdown of *NPF1a* or *NPF2* in S-phase locusts (n = 3 samples, one-way ANOVA, p<0.05). (G) NOS activity after injection of NPF1a or NPF2 peptide into G-phase locusts (n = 4 samples, 12–16 locusts/sample, one-way ANOVA, p<0.05). (H) NO levels after injection of NPF1a or NPF2 peptide into G-phase locusts (n = 4 samples, 12–16 locusts/sample, one-way ANOVA, p<0.05). (I) NOS activity 48 hr after transcript knockdown of *NPF1a* or *NPF2* in S-phase locusts (n = 4 samples, one-way ANOVA, p<0.05).

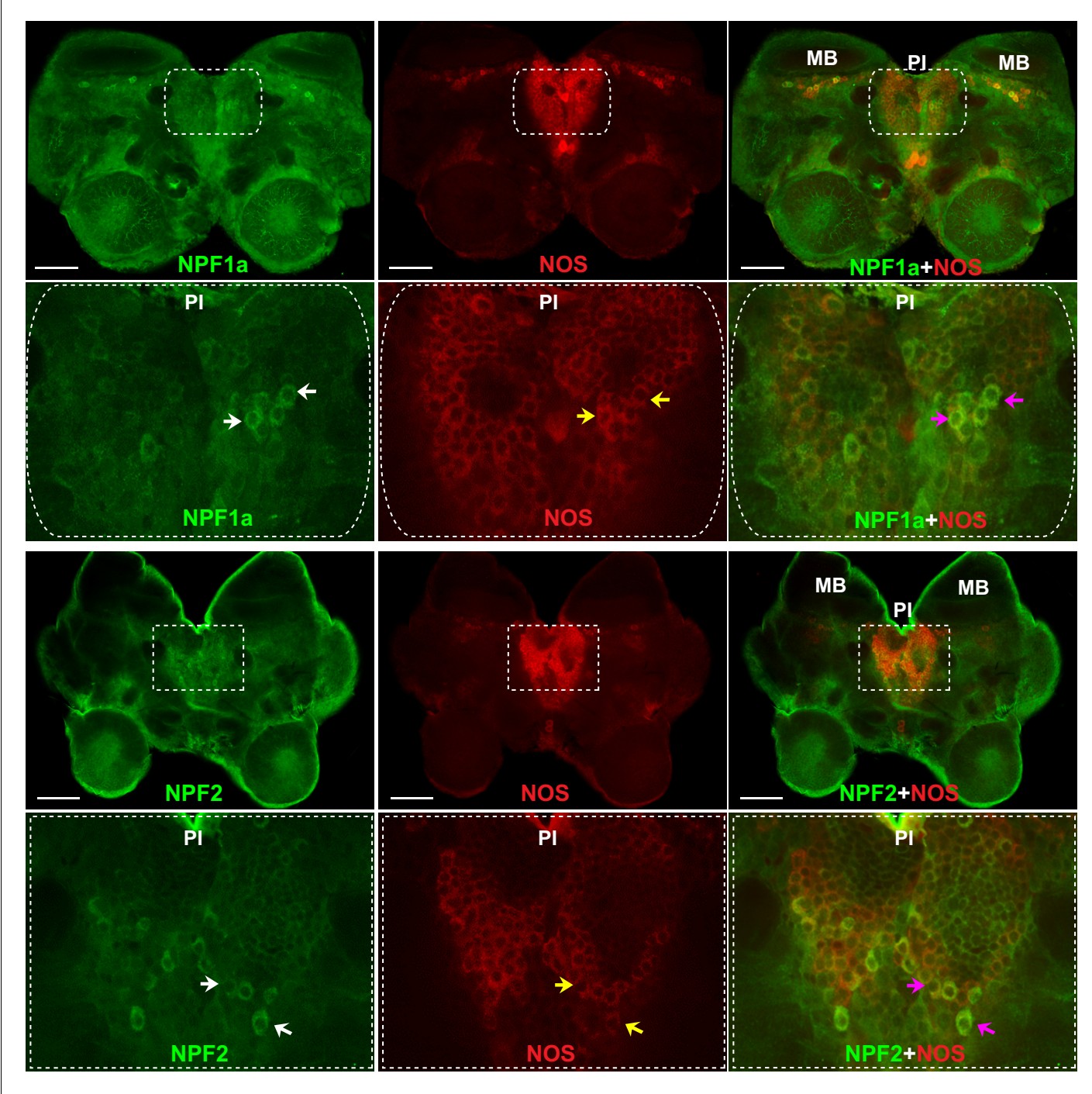

**Figure 8.** NOS and the two neuropeptides NPF1a and NPF2 co-localize in the pars intercerebralis of the locust brain. NPF1a and NOS also co-localize in the neurons of pars lateralis anterior to the calyces of mushroom in each hemisphere in the locust brain (see *Figure 8—figure supplement 1*). White arrows indicateNPF1a or NPF2 staining, yellow arrows show NOS staining, pink arrows indicate merged signal of NOS and NPF1a or NOS and NPF2. Scale bars represent 100 μm.

The following figure supplement is available for figure 8:

**Figure supplement 1.** NOS and the two neuropeptides NPF1a and NPF2 co-localize in the pars lateralis around the mushroom bodies in each hemisphere of locust brain.

cell body of neurons in the pars intercerebralis (*Figure 8*, lower). The co-localization of NPF1a and NPF2 with NOS in the pars intercerebralis of locust brain supports their linked action in phase-related behavioral changes.

### NPFR and NPYR separately mediate distinct regulatory mechanisms involving NPF1a and NPF2 on NOS phosphorylation and transcription

On the basis of the different binding activities between each NPF and the two receptors, we speculated that the two NPF receptors, *NPFR* and *NPYR*, are responsible for the distinct effects on NOS induced by NPF1a and NPF2 (phosphorylated NOS levels were decreased by NPF1a injection whereas *NOS* transcript levels were reduced by NPF2 injection, as shown in *Figure 7B,D*). Knockdown of the *NPFR* transcript in S-phase locusts increased NOS phosphorylation level without affecting *NOS* transcript level (*Figure 9A,B*), similar to the effect caused by *NPF1a* knockdown (*Figure 7A,D*). By contrast, knockdown of the *NPYR* transcript led to increased *NOS* mRNA and NOS protein levels (*Figure 9C,D*). Furthermore, we investigated whether NPF1a and NPF2 could affect NOS phosphorylation or transcript level in G-phase locusts in which the transcripts of *NPFR* or *NPYR* had been knocked down. We found that knockdown of the *NPFR* transcript relieved the inhibition of NOS phosphorylation caused by NPF1a administration (*Figure 9E,F*), whereas knockdown of the *NPYR* transcript blocked NPF2-induced reduction in *NOS* mRNA and NOS protein levels in G-phase locusts (*Figure 9G,H*). These data indicate that NPFR and NPYR mediate distinct effects of NPF1a and NPF2 on NOS phosphorylation and transcription, respectively, in the locust brain.

### NO levels mediate the effects of NPF1a/NPFR and NPF2/NPYR on locomotor behavior related to phase transition

To determine whether the NPF-induced NO reduction directly regulates phase-related locomotor plasticity, we conducted rescue experiments by administrating SNAP to enhance NO concentration in G-phase locusts pre-treated with NPF1a or NPF2 peptide. SNAP administration resulted in robust recovery of the $P_{greg}$ values, total duration of movement, and total distance moved for G-phase locusts in which $P_{greg}$ values had been reduced by injection of either NPF1a or NPF2 peptide (*Figure 10A*).

We then tested the effects of the NOS inhibitor L-NAME in S-phase locusts that had been pre-treated with ds*NPFR* or ds*NPYR*. Transcript knockdown of either *NPFR* or *NPYR* enhanced phase-related locomotor activity and thus promoted the behavioral shift from S-phase state towards G-phase state (*Figure 10B*). However, L-NAME administration robustly abolished the increase in $P_{greg}$ values, total duration of movement, and total distance moved for test locusts induced by *NPFR* or *NPYR* transcript knockdown. These data suggest that NO signaling is an essential mediator for the effects of two NPFs and their receptors on phase-related locomotor plasticity in locusts.

## Discussion

The current study reveals the inhibitory roles of two related neuropeptides, NPF1a and NPF2, and their receptors, NPFR and NPYR, in the locomotor activity related to locust phase transition. We provide evidence that NOS/NO signaling is a major mediator that transmits the effects of two NPF systems on phase-related locomotion. We establish a causation — the transcriptional changes in two NPF systems and the resulting converse alteration in NO levels in the locust brains contribute to variable locomotor activity during the G/S locust phase transition. Remarkably, NPF1a/NPFR and NPF2/NPYR suppress NOS activity and NO concentration at the levels of post-translational modification and transcription, respectively (see model in *Figure 11*).

### The NPF/NO signaling pathway plays an essential role in phase-related locomotor plasticity in locusts

We show that manipulating the levels of two NPFs by peptide injection or transcript knockdown significantly affects phase-related behaviors among four neuropeptides that had differential levels during locust phase transition. These changes in locomotor behavior can be fully overcome by pharmacological administration of compounds that affect NO levels. Notably, NO signaling displays marked effects on locomotor activity; and the time-course changes in NO levels coincide well with

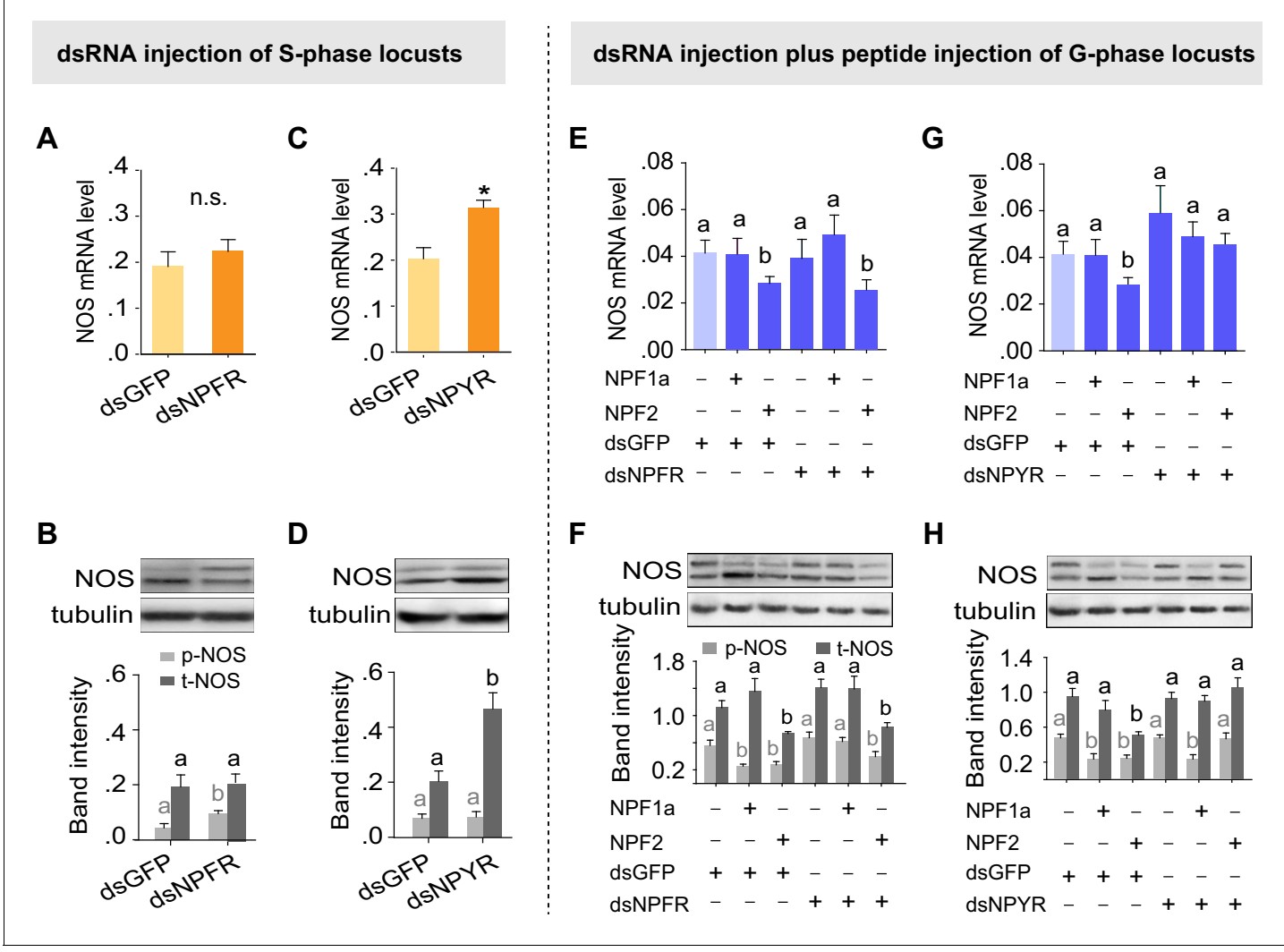

**Figure 9.** Two receptors mediate distinct effects of NPF1a and NPF2 neuropeptides on NOS phosphorylation and on *NOS* transcript levels, respectively. (**A**) and (**B**) *NOS* mRNA levels (n = 5 samples, 6–8 locusts/sample) and NOS protein levels (n = 3 samples, 10–12 locusts/sample) 48 hr after transcript knockdown of *NPFR* in S-phase locusts. The data are presented as mean ± s.e.m. Significant differences are denoted by letters. n.s. means not significant (Student's *t*-test, *p<0.05). (**C**) and (**D**) *NOS* mRNA levels (n = 5 samples) and NOS protein levels (n = 3 samples) 48 hr after transcript knockdown of *NPYR* in S-phase locusts. (**E**) and (**F**) *NOS* mRNA levels (n = 4 samples) and NOS protein levels (n = 3 samples) 4 hr after injection of NPF1a or NPF2 peptide in G-phase locusts pre-injected with ds*NPFR*. (**G**) and (**H**) *NOS* mRNA levels (n = 4 samples) and NOS protein levels (n = 3 samples) 4 hr after injection of NPF1a or NPF2 peptide in G-phase locusts pre-injected with ds*NPYR*. Detailed data describing NOS expression after injection of NPF1a or NPF2 peptide in G-phase locusts pre-injected with ds*NPFR* or ds*NPYR* are shown in *Figure 9—source data 1*.

The following source data is available for figure 9:

**Source data 1.** *NOS* mRNA levels and NOS protein levels after injection of NPF1a or NPF2 peptide into G-phase locusts pre-injected with dsNPFR or dsNPYR.

locust behavioral transitions during both isolation and crowding (*Guo et al., 2011*), indicating NO is a decisive molecule for phase-related locomotion. The increased NO concentration may serve as a proximate cause of high locomotor activity in G-phase locusts, and decreased NO levels may lead to low locomotor activity in S-phase locusts. These data clearly suggest that the NPF/NO pathway plays a vital role in the modulation of phase-related locomotor plasticity. Our studies do not, however, preclude the regulatory roles of two other neuropeptides, ACP and ILP, in other phase-related characteristics or in long-term behavioral effects (*Pener and Simpson, 2009*).

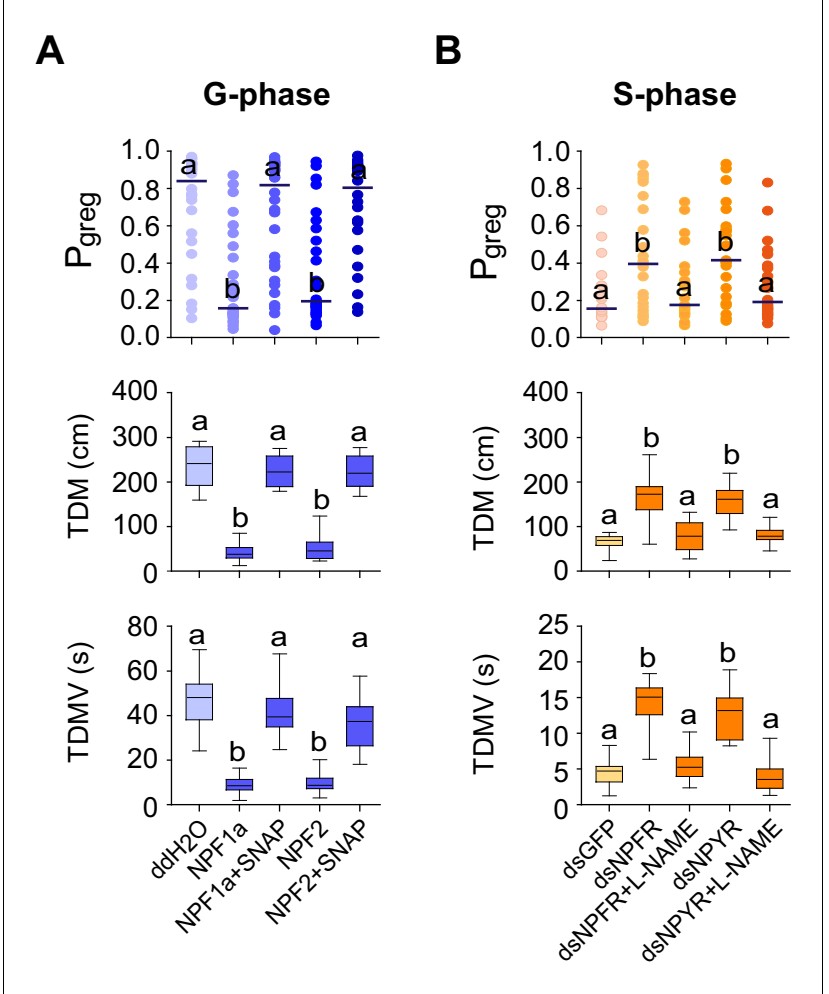

**Figure 10.** NPF1a, NPF2 and compounds that affect NO levels mediate effects on the locomotor behavior of G-phase and S-phase locusts. (**A**) Behavioral test after administration of NO donor (SNAP) to G-phase locusts pre-injected with NPF1a or NPF2 peptide. Significant differences are denoted by letters. For $P_{greg}$ analysis, lines indicate median value (n $\geq$ 24 locusts; Mann–Whitney U test, p=0.0003 and 0.0001 for $P_{greg\ NPF1a\&SNAP}$ vs. $P_{greg\ NPF1a}$ and $P_{greg\ NPF2\&SNAP}$ vs. $P_{greg\ NPF2}$, respectively). For TDM and TDMV analysis, the data are presented as mean ± s.e.m. (n $\geq$ 24 locusts, Student's *t*-test, p<0.05). (**B**) Behavioral test after administration of NOS inhibitor (L-NAME) in S-phase locusts pre-injected with ds*NPFR* or ds*NPYR* (n $\geq$ 16 locusts, Mann–Whitney U test, p=0.022 and 0.042 for $P_{greg\ dsNPFR\&L-NAME}$ vs. $P_{greg\ dsNPFR}$ and $P_{greg\ dsNPYR\&L-NAME}$ vs. $P_{greg\ dsNPYR}$, respectively). For TDM and TDMV analysis, the data are presented as mean ± s.e.m. (n $\geq$ 16 locusts, Student's *t*-test, p<0.05).

Numerous studies have suggested that NPF signaling can influence a broad range of physiological and behavioral activities in insects, for instance feeding, reproduction, learning, circadian activity and stress responses (*Nässel and Wegener, 2011*; *Lee et al., 2006*; *Krashes et al., 2009*). Similar functional roles of the NPF (or NPFY) system in locomotion have been observed in two model invertebrate species, *Caenorhabditis elegans* (*de Bono and Bargmann, 1998*) and *Drosophila melanogaster* (*Wu et al., 2003*). These findings, in conjunction with ours, raise the possibility that the NPF system might serve as a common neural signaling pathway that shapes locomotor plasticity in invertebrates. In addition, recent studies have reported that NPF (referred to as NPF1a here) can mediate food intake, body weight and male-specific reproduction processes in the adults of another locust species, *Schistocerca gregaria* (*Van Wielendaele et al., 2013a*, *2013b*), indicating that NPF plays multiple roles in locust biology.

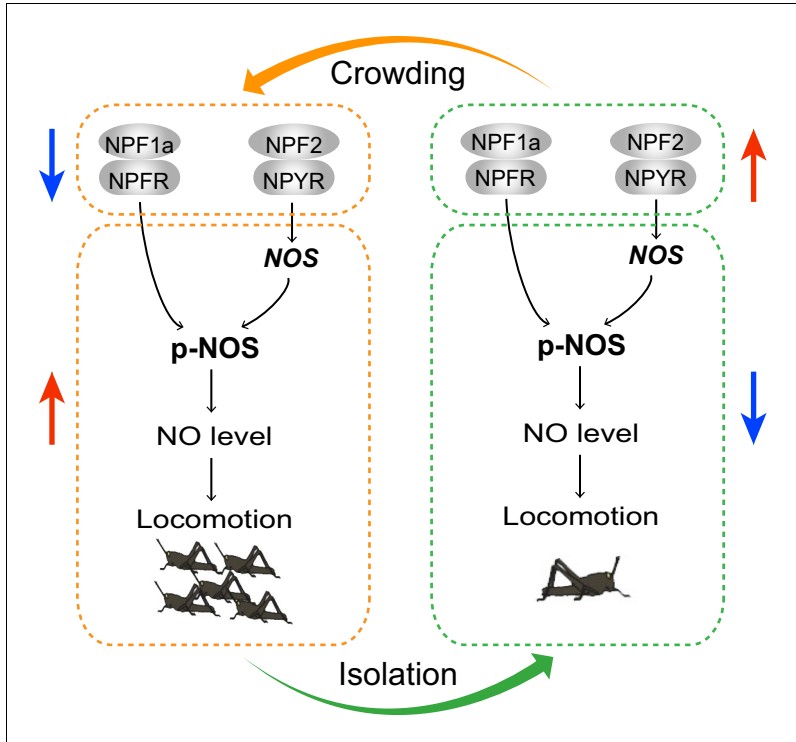

**Figure 11.** A model for the effects of neuropeptides NPF1a and NPF2 on locomotor activity related to phase transition of the migratory locust through NO signaling. During the G/S phase transition, changes in the expression of two NPFs and their receptors act in concert to regulate the NO level, thus shaping phase-related locomotor plasticity. During crowding, the levels of two NPF systems decrease and promote NO level, resulting in higher locomotor activity. During isolation, the levels of two NPF systems increase and reduce NO level, thus leading to lower locomotor activity. Arrows indicate increased or decreased levels or activity.

Previous studies have suggested that phase differences in food choice are related to cryptic and aposematic strategies. Gregarious nymphs of *S. gregaria* are prone to consume more nutritionally imbalanced food and to accept food containing toxic chemicals more readily than do solitarious nymphs (*Simpson et al., 2002*). This is partly due to the relative gustatory insensitivity to low-quality food in gregarious locusts (*Despland and Simpson, 2005*). A link between NPF signaling and food choice has been suggested in *Drosophila* (*Shen and Cai, 2001*). Thus, a possible involvement of the NPF system could be speculated in phase-related behavioral choices of food quality in the locust nymphs. Further functional analysis is required to confirm this possibility and to elucidate the mechanisms through which NPF regulates multiple phase-related behavioral characteristics.

Our results indicate that NO signaling has a stimulatory role in locomotor activity in locusts. Several studies have suggested the significance of NO signaling in locomotor activity and behavioral plasticity in various animal species (*Del Bel et al., 2005*; *Kyriakatos et al., 2009*). For example, NO-initiating signaling has been shown to suppress aggression by promoting the tendency to flee in crickets (*Stevenson PA, 2016*) and to increase oviposition digging rhythm so as to control egg-laying movements in desert locusts (*Newland and Yates, 2007*). The cGMP/protein kinase G (PKG) pathway (the main downstream target of NO signaling) is involved in the control of foraging and locomotor behavior in *Drosophila* (*Osborne et al., 1997*), as well as in the regulation of labor division in honey bees (*Ben-Shahar, 2005*) and ants (*Ingram et al., 2005*). However, although there is a phase-dependent regulation of NO synthesis (*Rogers et al., 2004*) and a higher PKG activity in the anterior midline of brains of insects in the gregarious phase (*Lucas et al., 2010*), significant effects of PKG on behavioral phase state could not be observed in the desert locust *S. gregaria* (*Ott et al., 2012*). A reasonable explanation could be that the regulatory mechanisms of NO signaling in phase transition are species-specific, as is true of the roles played by several other neurotransmitters

(*Ma et al., 2011*; *Anstey et al., 2009*). Another possibility is that NO regulates behavioral phase transition via a PKG-independent pathway in locusts (*Newland and Yates, 2008*).

## NPF1a and NPF2, their receptors and NOS act in concert to regulate phase-related locomotion through NO signaling

We provide clear evidence that two NPFs acts as brakes that sequentially modify NO levels to control locomotor plasticity. The regulatory role of the NPF-NO pathway in locomotor behavior is further supported by the overlap immunostaining of two NPFs and NOS in the pars intercerebralis, which is linked to the regulation of locomotor rhythm in insects (*Matsui et al., 2008*). NO levels may reflect distinct physiological states and affect a wide variety of behaviors across species (*Collmann et al., 2004*; *Davies, 2000*; *Del Bel et al. 2005*; *Cayre et al., 2005*), yet how this molecule's level responds to varied internal or external conditions remains unclear. To the best of our knowledge, this study is the first to show the link between NPF and NO signaling in shaping behavioral plasticity.

We show that the sequential inhibitory effects of NPF1a and NPF2 on NO levels are attributed to their regulation of NOS phosphorylation and *NOS* gene transcription, respectively, indicating that these two NPF members are not redundant in regulating phase-related locomotion. Phosphorylation is known to be an important form of post-translational modification (PTM) for a broad range of proteins, including receptors, transcriptional factors and vital enzymes (*Kasuga et al., 1982*; *Matsuzaki et al., 2003*; *Bertorello et al., 1991*). The phosphorylated proteins usually display changed spatial structures, subcellular locations and catalytic activity, and thus play key roles in rapid cellular signaling (*Aguirre et al., 2002*; *Ho et al., 2011*; *Hurley et al., 1990*). Studies in mammals have shown that NOS activity is tightly regulated by phosphorylation. For instance, the phosphorylation of Ser1412 stimulates NOS activity whereas Ser847 phosphorylation inhibits enzyme activity (*Watts et al., 2013*; *Komeima et al., 2000*).

NOS has also been suggested to be modified post-translationally in the locust embryo (*Stern et al., 2010*). Here, we show that NOS is modified by phosphorylation in the locust brains. Even if the total NOS protein level were not influenced by the activities of NPFs, simply reducing NOS phosphorylation leads to significantly decreased NOS activity and thus results in lower NO level, suggesting that NOS activity in the locust largely depends on its modification by phosphorylation. Therefore, NPF1a may lower the NO level by directly reducing NOS phosphorylation, whereas NPF2 may lower the NO level by reducing NOS substrate for phosphorylation. Our results show that NPF1a-regulated NOS phosphorylation cycles quickly, whereas *NOS* expression may respond more slowly to NPF2 regulation. Thus, the distinct modes of changing NO levels that are regulated by the two NPF systems not only explain the more rapid behavioral effect of NPF1a when compared to that of NPF2, but also emphasizes that downregulation of the NPF2 system is necessary in the G-phase to increase locomotion.

We show that two NPF peptides and their receptors may play synergistic roles in regulating the dynamic changes in NO levels during the two time-course processes of phase transition. The continuous reduction of NO levels during isolation is tightly controlled by the decreased NOS phosphorylation that results from the upregulation of *NPFR* and *NPYR*. By contrast, the reduction of two *NPF*s contributes mainly to the overall enhancement of NOS phosphorylation and NO levels during crowding. Although phosphorylated NOS shows greater activity than the unphosphorylated protein in promoting NO production, as shown previously, the enhancement of NO levels upon crowding seems to be delayed relative to that of NOS phosphorylation, implying that the stimulation of NO levels during crowding is a complex process that might involve additional regulators beyond the enzyme activity. NO level is dependent upon the balance between its production and degradation (*Sansbury and Hill, 2014*). NO generation not only depends on NOS expression and its post-translational modification but also relies on the availability of the corresponding substrate (e.g., L-Arginine) and cofactor (e.g., BH4, FAD or FMN) (*Li and Poulos, 2005*), whereas NO degradation may result directly from its reaction with reactive species (e.g., superoxide) (*Channon, 2012*). Given this, modulations of the availability of these factors may responsible for the sluggish increase of NO level during locust crowding.

## Specific effects of different neuromodulators are essential for orchestrating phase-related behavioral traits

Locomotor activity is a major phase-related behavior that changes in response to population density (*Wang and Kang, 2014*). The high locomotor activity of G-phase locusts is potentially beneficial for rapid aggregation, synchronous movement, and avoidance of predators or conspecific cannibalism during locust swarming (*Simpson et al., 1999*). Therefore, the sequential modifications of NO levels resulting from NPF1a and NPF2 should allow dynamic locomotor adaptation to maintain locust swarming. Our previous studies have indicated that several other regulators, such as dopamine, serotonin and carnitines, are also involved in the modulation of phase-related locomotion in the migratory locust (*Wu, et al., 2012*; *Ma et al., 2015*). In addition, protein kinase A, a possible downstream factor of serotonin and dopamine, can regulate behavioral phase transition in the desert locust (*Ott et al., 2012*). It has been shown that the NPF/NPFR pathway has a dominant suppressive effect on PKA-sensitized sugar aversion in *Drosophila* (*Xu et al., 2010*). In our study, the expression level of AC2, one of the enzymes catalyzing cAMP production and activating PKA, is also affected by alteration of NPF levels in locusts. Studies in mammals have shown that both dopaminergic transmission and PKA could enhance NO levels thus leading to distinct biological actions (*Wang and Lau, 2001*; *Yang et al., 2011*). On the basis of these findings, we hypothesize that the two NPF systems may cooperate with the dopamine pathway to modulate locomotor activity during locust phase transition.

We show that the NPF/NO pathway is not involved in the modulation of another major phase-related behavioral characteristic, conspecific attraction induced by odors, in the migratory locust. This finding is superficially inconsistent with previous results on the roles of NPFs or NO in fine-tuning of food odor-induced behavior and olfactory learning in mice and the fruit fly (*Rohwedder et al., 2015*; *Sung et al., 2014*). One possible explanation is that pheromone-induced olfactory behaviors that are related to the locust phase change may involve regulatory mechanisms that are different from those involved in food-odor-induced olfactory responses in locusts. And, the locust phase transition is a continuous process involving changes of various characteristics including behaviors, metabolism, immunity and body color (*Wang and Kang, 2014*). In addition to its significance in behavioral modulation, NO signaling is also able to affect a variety of physiological and pathological processes (*Bogdan, 2015*; *Calabrese et al., 2004*; *Sansbury and Hill, 2014*). Thus, uncovering the long-term effects of the NPF/NO pathway on phase-related characteristics, such as disease resistance, energy metabolism and aging, will provide a more comprehensive understanding of the phase transitions that underlie locust swarming.

## Materials and methods

### Rearing of locusts

G-phase locusts were maintained in large well-ventilated cages (40 cm × 40 cm × 40 cm) at a density of 500–1000 locusts per cage. S-phase locusts were reared individually in boxes (10 cm × 10 cm × 25 cm) supplied with charcoal-filtered compressed air. Both colonies were maintained at 30 ± 2°C and under 14:10 light/dark photocycle regime. The locusts were fed with fresh wheat seedling and bran (*Guo et al., 2011*).

### Experimental samples for time-course analysis of gene expression during phase transition

For solitarization, fourth-instar G-phase nymphs were separately raised under solitarious conditions as described above. After 0, 1, 4, 16, or 32 hr of isolation, locust brains were collected and snap frozen. For gregarization, two fourth-instar S-phase nymphs were reared in small cage (10 cm × 10 cm × 10 cm) containing 20 G-phase locusts of the same developmental stage. After 0, 1, 4, 16, or 32 hr of crowding, locust brains were dissected and frozen in liquid nitrogen. All samples were stored at −80°C. Each sample contained a total of eight insects, including four male and four female insects. Four independent biological replicates were prepared for further experiments.

## RNA preparation and qPCR

Total RNA was extracted using the RNeasy Mini Kit (Qiagen) according to the manufacturer's protocol. cDNA was reverse-transcribed from 2 μg of total RNA using M-MLV reverse transcriptase (Promega). Gene-specific mRNA levels were assessed by qPCR using the SYBR Green kit on a LightCycler 480 instrument (Roche). *RP49* was used as internal reference. The primers used are shown in *Supplementary file 2*.

## Transcript knockdown via RNAi

The dsRNAs of target genes were prepared using the T7 RiboMAX Express RNAi system (Premega). ds*RNA* was microinjected into the brains of test insects (69 ng/locust for *NPF1a*, *NPF2*, *ACP*, *ILP*, and *NOS*, 1 μg/locust for *NPFR* and *NPYR*). ds*GFP-RNA* was used as control in all RNAi experiments. The behaviors of test locusts were measured 48 hr after injection as described below.

## Peptide injection and drug treatments

The concentrations of peptides and drugs that were used were determined according to described methods (*Newland and Yates, 2007*; *Van Wielendaele et al., 2013b*). The commercially synthesized peptides (BGI, NPF1a peptide — YSQVARPRF-NH$_2$; and NPF2 peptide — RPERPPMFTSPEE LRNYLTQLSDFYASLGRPRF-NH$_2$) were dissolved in ddH$_2$O as stock solution (20 μg/μl). Working solutions of different concentrations (0.05, 0.5 and 2.5 μg/μl) were injected into the hemolymph in the heads of fourth-instar locusts using a microinjector (2 μl/locust). The arena behavioral assay was conducted 1 hr, 2 hr, and 4 hr following injection. L-NAME was dissolved with ddH$_2$O to make a 1 mM stock solution. Working solution (100 μM) was microinjected into the heads of G-phase locusts (2 μl/ locust). SNAP was dissolved with DMSO to prepare 100 mM stock solution. Working solution (200 μM) was microinjected into the heads of S-phase locusts (2 μl/locust). Locust behaviors were tested 2 hr after drug injection.

## Behavioral arena assay

The behavioral assay was performed in a rectangular Perspex arena (40 cm $\times$ 30 cm $\times$ 10 cm) containing three chambers. The left chamber (7.5 cm $\times$ 30 cm $\times$ 10 cm) contained 30 fourth-instar G-phase locusts as a stimulus group, and the right chamber was empty (7.5 cm $\times$ 30 cm $\times$ 10 cm). Locusts behaviors were recorded for 300 s by an EthoVision video tracking system and analyzed according to the binary logistic regression model constructed in our previous work (*Guo et al., 2011*). Details are as follow: P$_{greg}$ = e$^{\eta}$/ (1+e$^{\eta}$); $\eta$ = −2.11 + 0.005 $\times$ AI (attraction index) + 0.012 $\times$ total distance moved +0.015 $\times$ total duration of movement; AI = total duration in stimulus area − total duration in the opposite of stimulus area; this parameter represents the extent to which the tested animals are attracted by the stimulus group. TDMV (total duration of movement) and TDM (total distance moved) indicate the locomotor activity levels. P$_{greg}$ indicates the probability that a locust is considered gregarious. P$_{greg}$ = 1 represents a fully gregarious behavior, whereas P$_{greg}$ = 0 represents a fully solitarious behavior. In the behavioral assay, 16–35 locusts were tested for each treatment according to the sample size reported in previous studies (*Ott et al., 2012*; *Ma et al., 2011*). Locusts that did not move during behavioral testing were excluded.

## Characterization of NPF receptors in locusts

The amino acid sequence of the *Drosophila* NPF receptor was used to search for NPF homologs in the locust genome database utilizing the tblastn algorithm. The phylogenetic relationship of NPFR and NPYR of insects and human was analyzed using MEGA software.

HEK 293 T cells (RRID: CVCL-0063) were purchased from the American Type Culture Collection (ATCC, CRL-3216, the identity has been authenticated using STR profiling) and cultured in low glucose DMEM (Life Technology) supplemented with 10% fetal bovine serum. Cells were routinely tested for mycoplasma every 6 months. For the competition binding assays, HEK 293 T cells transiently transfected with pcDNA3.1-NPFR or pcDNA3.1-NPYR (with a Flag-tag encoding sequence following target gene) were washed with 1 X PBS and added into 96-well plates (2 $\times$ 10$^5$ cells/well) coated with poly-L-lysine (0.1 mg/mL). Cells were then incubated with 25 μL TAMRA-NPF1a or TAMRA-NPF2 (10 nM) in the presence of increasing concentrations of unlabeled ligands in a final volume of 100 μL of binding buffer (PBS containing 0.1% bovine serum albumin). Nonspecific

binding was determined by the addition of 25 µL unlabeled ligand. Mixtures were incubated at 30°C for 2 hr. Fluorescence intensity was measured using a fluorimeter (Molecular Devices) after washing twice with binding buffer. The HEK 293 T cells transfected with pcDNA3.1 were used as a control. The binding displacement curves were analyzed using the non-linear logistic regression method. Western blotting was carried out to validate the protein expressions of NPFR and NPYR in HEK 293 T cells using the mouse monoclonal antibody against Flag (CoWin, 1:5000).

## RNA-seq and data processing

The brains of fourth-instar G-phase locusts were collected 4 hr after injection of the mixture of NPF1a and NPF2 peptides or ddH$_2$O (a total of 5 µg). Similarly, the brains of fourth-instar S-phase locusts were collected 48 hr after injection of the mixture of ds*NPF1a* and ds*NPF2* or ds*GFP*. Each sample contained 10 brains (5 males and 5 females). Three independent replicates were performed for each treatment. Total RNA was isolated as previously described, and RNA quality was confirmed by agarose gel. cDNA libraries were prepared according to Illumina's protocols. Raw data were filtered, corrected, and mapped to locust genome sequence using Tophat software. The number of total reads was normalized by multiple normalization factors. Transcript levels were calculated using the reads per kb million mapped reads criteria. The difference sbetween the test and control groups were represented by *P* values. Differentially expressed genes with significance levels at p<0.05 in each comparison were enriched. In addition, unsupervised hierarchical clustering was performed using Clustal 3.0, which employs uncentered Pearson correlation and average linkage; results are presented by Java Treeview software. The RNA-seq data have been deposited in the Sequence Read Archive database of the National Center for Biotechnology Information (NCBI) (accession no. SRP092214).

## Western blot analysis

Locust brains (10–12 individuals/sample) were collected and homogenized in 1 X PBS buffer (0.1 M phosphate buffer, 0.15 M NaCl, pH 7.4) containing the phosphatase inhibitor PhosSTOP (Roche) and a proteinase inhibitor (CoWin). Total protein content was examined using the BicinChoninic Acid (BCA) Protein Assay Kit (Thermo). The extracts (100 µg) were reduced, denatured, and electrophoresed on 8% SDS-PAGE gel and then transferred to polyvinylidene difluoride membrane (Millipore). The membrane was then cut to two pieces and incubated separately with a specific antibody against the target protein of ~130 KD or a reference protein of ~55 KD overnight at 4°C (affinity-purified polyclonal rabbit antibody against uNOS, Sigma-Aldrich, 1:200; Rabbit polyclonal antibody against tubulin, CoWin, 1:5000). Goat anti-rabbit IgG was used as secondary antibody (CoWin, 1:10000). Protein bands were detected by chemiluminescence (ECL kit, CoWin). The band intensity of the Western blot was quantified using densitometry in Quantity One software.

For determination of NOS phosphorylation, 200 µg brain extracts were incubated with λ phosphatase and 1 X NEB buffer supplemented with 1 mM Mncl$_2$ for 1 hr at 30°C (NEB). Control protein was treated under the same conditions without λ phosphatase. Western blot analysis was performed to confirm NOS phosphorylation in the locust brains.

## Bioassays

An enzyme-linked immunosorbent assay (ELISA) kit (R and D Systems, Inc.) was used to detect the relative cAMP level in locust brains. For NO content determination, Total Nitric Oxide Assay Kit (Beyotime) was used. Because NO molecules are unstable, the total NO levels in all test groups were assessed by detecting the content of nitrate and nitrite. The NOS Kit (Nanjing Jiancheng Bioengineering Institute) was used to detect the total NOS activity in locust brains. Protein concentrations were measured using the BicinChoninic Acid (BCA) Protein Assay Kit (Thermo). All of these three measurements were performed according to the manufacturer's instructions. Each measurement was from at least four biological replicates (12–16 locusts/replicate). Data were normalized to the protein concentration.

## Immunohistochemistry

Whole-mount double immunohistochemistry of locust brains was performed using affinity-purified polyclonal rabbit antibody against NPF1a or NPF2 (AbMAX, China, 1:100) and monoclonal mouse

antibody against uNOS (Thermo, 1:200, RRID: AB_325476). Alexa Fluor-488 goat anti-rabbit IgG (Cat. A-11008, 1:500; Life Technologies) and Alexa Fluor-568 goat anti-mouse IgG (Cat. A-11019, 1:1000; Life Technologies) were used as secondary antibodies for NPFs and NOS staining, respectively. Fluorescence was detected using an LSM 710 confocal laser-scanning microscope (Zeiss). Photos for both positive staining and negative controls were imaged under the same conditions.

## Determination of the molecular effects on NOS expression caused by NPFR and NPYR

To validate the involvement of *NPFR* and *NPYR* in the regulation of NOS expression and phosphorylation by two NPF peptides, the brains of fourth-instar S-phase locusts were microinjected with ds*NPFR* or ds*NPYR*, and collected 48 hr after injection. For gregaria, the brains of fourth-instar locusts were microinjected with ds*NPFR*, ds*NPYR* or ds*GFP* followed by NPF1a or NPF2 treatment 4 hr before sample collection. Total RNA and protein in each treatment were extracted according to the Invitrogen TRIzol RNA and protein extraction protocol. qPCR and Western blot analysis were performed to examine the influence of NPFR and NPYR on NOS expression and phosphorylation.

## Behavioral rescue experiments in vivo

For G-phase locusts, synthesized NPF1a or NPF2 peptide (2.5 µg/µl) was microinjected into the fourth-instar insects. Two hours later, SNAP (200 µM, 2 µl/locust) was injected into the heads of the experimental insects. Control insects were treated with an equal amount of saline. The injected locusts were then raised under the gregarious condition and subjected to behavioral analysis 2 hr after injection of SNAP.

For S-phase locusts, ds*NPFR* or ds*NPYR* was microinjected into the brains of fourth-instar S-phase insects. Forty-six hours after injection, the NOS inhibitor L-NAME was microinjected into the locusts pre-treated with ds*NPFR* or ds*NPYR*. The insects treated with ds*GFP* were used as a control. Tested insects were thus raised under the solitarious condition and subjected to behavioral analysis 2 hr after injection of L-NAME.

## Statistical analyses

For gene expression and enzyme activity analysis, we knew from the previous studies that a sample size of 6 animals per treatment was enough to detect significant differences among treatments (*Ott et al., 2012*; *Yang et al., 2014*). Therefore, 8–16 animals were examined in each experimental treatment. For behavioral measurement, we knew that 15 individuals per group was sufficient to detect reproducible differences between groups (*Ma et al., 2011*). All of the experiments were performed with at least three independent biological replicates.

Student's *t*-test was used for two-group comparison. One-way ANOVO followed by Turkey's post-hoc test was used for multi-group comparisons. Data that do not meet normal distribution were excluded in these statistics. Behavioral phase state analysis was performed using the Mann–Whitney U test because of its non-normal distribution feature. Differences were considered statistically significant at $p < 0.05$. Data were analyzed using SPSS 20 software and presented as mean ± s. e.m. except for the $P_{greg}$ values, which are shown as median values.

## Acknowledgements

We thank Gerald Reeck (Kansas State University and Institute of Zoology, Chinese Academy of Sciences) for constructive comments during the revision of this manuscript. We are grateful to Prof. Minmin Luo (National Institute of Biological Sciences, Beijing Institute), Prof. Chuan Zhou (Institute of Zoology, Chinese Academy of Sciences) for their insightful suggestions on this project. This work was supported by the Strategic Priority Research Program of CAS (Grant NO. XDB11010000) and the National Natural Science Foundation of China (Grant NO. 31601875 and 31472047).

## Additional information

### Funding

| Funder | Grant reference number | Author |
| --- | --- | --- |
| National Natural Science Foundation of China | Youth fund (Grant NO. 31601875) | Li Hou |
| National Natural Science Foundation of China | Grant NO. 31472047 | Xianhui Wang |
| Chinese Academy of Sciences | Strategic Priority Research Program (Grant NO. XDB11010000) | Xianhui Wang Le Kang |

The funders had no role in study design, data collection and interpretation, or the decision to submit the work for publication.

### Author contributions

LH, Data curation, Funding acquisition, Writing—original draft, Writing—review and editing; PY, Data curation, Methodology, Analysis and interpretation of data; FJ, Analysis of the gene sequences; QL, Gene cloning; XW, Conceptualization, Funding acquisition, Writing—original draft, Writing—review and editing; LK, Conceptualization, Funding acquisition, Writing—original draft, Project administration, Writing—review and editing

### Author ORCIDs

Li Hou, http://orcid.org/0000-0001-6727-7053
Le Kang, http://orcid.org/0000-0003-4262-2329

## Additional files

### Supplementary files

• Supplementary file 1. Protein sequences of two NPF receptors, NPFR and NPYR, in the migratory locust.

• Supplementary file 2. Primers used in qPCR and RNAi experiments.

### Major datasets

The following dataset was generated:

| Author(s) | Year | Dataset title | Dataset URL | Database, license, and accessibility information |
| --- | --- | --- | --- | --- |
| Yang PC | 2016 | Locusta migratoria transcriptome | https://www.ncbi.nlm.nih.gov/sra/?term=SRP092214 | Publicly available at the NCBI Sequence Read Archive (accession no: SRP092214) |

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
