## [Decision Letter]

Thank you for submitting your article "Nitric oxide mediates the effects of neuropeptides NPF1a and NPF2 on locomotor plasticity underlying locust swarming" for consideration by *eLife*. Your article has been reviewed by three peer reviewers, and the evaluation has been overseen by K VijayRaghavan as the Reviewing and Senior Editor. The following individuals involved in review of your submission have agreed to reveal their identity: Shannon B Olsson (Reviewer #1); Amir Ayali (Reviewer #3).

The reviewers have discussed the reviews with one another and the Reviewing Editor has drafted this decision to help you prepare a revised submission.

Summary:

In this interesting study, the authors found that nitric oxide, regulated by NPF peptide mediation of nitric oxide synthase in the pars intercerebralis portion of the brain, leads to increased locomotor activity associated with the solitary to gregarious phase change in the migratory locust. The authors conclude that this pathway could be a key modulator underlying swarming behavior.

The paper is generally well-written, and covers a large body of work. The methods are thoroughly described, and the results well-presented – with an appropriate separation of results into the main paper and supplementary methods. In general the discussion and conclusions are well-supported by the results.

Essential revisions:

Reviewer #1:

The methodology and controls are thorough and the results regarding the impact of NPF peptides on NOS and NO are clear. However, it is unclear from the behavior presented whether the decreased/increased movement observed is directly related to swarming behavior, or rather a general increase in mobility. Obviously, increased mobility is needed for swarming, but the two are separate phenotypes. For example, increased locomotion would also affect a number of other behaviors, from foraging to mating, even though the authors show that, for example, it doesn't relate to changes in olfactory attraction to odors.

Second, the results, in my interpretation, are presented as though the NPF / NO pathways observed lead to the phase change, while they may rather be a byproduct of the transition itself. This issue could be alleviated by a deep dive into the literature, including discussion of previous publications by the authors (some of which are already included in the references). For example:

1) How might the molecular mechanisms presented relate to the many other physiological and behavioral changes that occur during phase change (Molecular Mechanisms of Phase Change in Locusts, Xianhui Wang and Le Kang, Annual Review of Entomology, Vol. 59: 225 -244)?

2) Nitric oxide is incredibly important for a number of neurophysiological processes, so how does this mechanism relate to other aspects of NO modulation?

3) How does this mechanism relate and/or differ from other observations of behavioral modulation by NO in orthopertans (Controlling the decision to fight or flee: the roles of biogenic amines and nitric oxide in the cricket Paul A. Stevenson and Jan Rillich, Current Zoology, 2016, 1-11), and other modulators in locusts observed by the authors previously (Modulation of behavioral phase changes of the migratory locust by the catecholamine metabolic pathway Zongyuan Ma, Wei Guo, Xiaojiao Guo, Xianhui Wang, and Le Kang, PNAS, 108, 3882-3887)?

4) Finally, how do these observations in l. migratoria relate to observations on the mechanisms of neuromodulation other locust species (Substantial changes in central nervous system neurotransmitters and neuromodulators accompany phase change in the locust, Stephen M. Rogers, Thomas Matheson, Ken Sasaki, Keith Kendrick, Stephen J. Simpson, Malcolm Burrows, Journal of Experimental Biology 2004 207: 3603-3617)?

These aspects of the results should be discussed to better interpret the results in light of current knowledge.

Reviewer #2:

My main criticism of this paper is that the Discussion focuses too much on reiterating the results, rather than focusing more on a synthetic interpretation of the results. This is reflected in the model presented in Figure 11, which provides little insight into how the neuropeptides, receptors and NOS enzyme act in concert to regulate locomotion through nitric oxide signalling. The model as presented suggests that p-NOS and NOS are equally active, and ignores much of the dynamics of the NPFs and receptors described in the results. To this reviewer, the results are consistent with the hypothesis that only p-NOS is active, and that NPF2 increases NOS activity only by providing a greater level of NOS substrate for phosphorylation. Also, that NPF1a-regulated NOS phosphorylation cycles quickly whereas NOS expression is slower to respond to NPF2 regulation not only explains the speed of locomotory transition between phases, but perhaps also explains why regulation of NPYR expression is necessary in the G-Phase to increase locomotion.

It would also help the reader if the effects of neuropeptide and NO signalling described in this paper could be put in context with the other hormonal and transcriptional regulators described previously. Where does the regulatory system described in this paper fit more broadly within the overall regulation of phase transition?

Reviewer #3:

This paper describes a very thorough investigation suggesting a potential role for NPF in locust density-dependent phase transition and in locust phase characteristics.

The paper is overall well written and the findings well described. I do have several comments on the study, the text and figures. The following (a mix of major and minor comments) is arranged by section:

Introduction

1) While this is not a focus of this work, the authors tend to use "swarming" and "collective motion" interchangeably, while these refer to different aspects of locust (and other animals) behavior. A related point is that the gregarious phase is not a characteristic of a swarming population, but is rather characterized by swarming behavior. This may seem as merely semantics, yet in a paper describing the control of phase-transformation, I believe it is important. The authors are referred to Ariel and Ayali, 2015, for a recent review addressing these points.

2) I am not sure that the sentence "Switching between key types of neuromodulators[…]" reflects the major points presented in the cited references, nor in the current text; the point being that neuromodulators are able of choosing among a behavioral repertoire and thus mediate behavioral plasticity. Similarly, neuropeptides induce behavioral plasticity rather than "modulate different types of behavioral plasticity".

3) The time course of a locust phase change is a repeated important aspect when reviewing the different presented results. It may deserve some further details and references (e.g. is gregarization indeed "initiates at least 32 h after crowding", or is it completed?).

4) While in the Abstract the authors state that the "neuropeptide F (NPF)-nitric oxide (NO) pathway plays a critical role in locomotor plasticity", in the Introduction they claim that the neuropeptides are crucial neuromodulators in "behavioral plasticity underlying swarming". As mentioned in my point 1 above, the connection between enhanced locomotion and swarming is not trivial.

Results

5) This is somewhat related to the issue of time courses mentioned earlier. Figure 1—figure supplement 1 shows 11 peptides' mRNA levels that do not change during phase transformation. According to this figure they also seem not to differ between the extreme phases. Yet, they were reported (and therefore presented here) as differentially expressed in gregarious and solitary locusts. Are these data contradicting? Should be explained/discussed. Furthermore, should one expect the levels of mRNA of peptides 0 hr after crowding to be similar to those of 32 hr after isolation and vice versa? This is not the case in Figure 1 or Figure 1—figure supplement 1 (nor in Figure 3, but compare to Figure 5). Last, based on Figure 1, an effect of crowding is seen more often than an effect of isolation and usually within 1-4 hr. How does this fact correspond with the claim that in L. migratoria crowding is very quick while isolation is slow? All this points need clarification/discussion.

6) The authors present two measures of behavioral phase (e.g. Figure 2): the one is Pgreg and the other distance/duration of movement. The reader may tend to take these as a measure of activity (the latter) and a measure of attraction to peers (the former). However, Pgreg is a result of a binary logistic regression model which includes or is heavily based on the distance/duration of movement! Are locusts injected with NPF only less active or also less attracted to peers? This point brings me back to the issue of an effect on swarming behavior vs. a mere effect on locomotion, which are not the same. Another measure appears in the supplementary figures – Attraction index. There is no definition or explanation for how this measure is calculated.

7) The most left columns (dsGFP) in Figure 2 seem to be similar. Why didn't crowding affect the control locusts (Figure 2)?

8) Not as described in subsection “Two NPF receptors, NPFR and NPYR, are essential for changes in locomotor activity related to phase transition”, according to Figure 3, the pattern of both NPFR and NPYR during isolation is very similar (increase) and somewhat similar during crowding (no change vs decrease).

9) Coming back yet again to the important question of time courses, the authors state in subsection “NO signaling is a vital modulator of locomotor plasticity in the G/S phase transition” that "The mRNA and protein levels of NOS […] showed strong correlation with the time course of G/S phase transition". This statement is not consistent with the Introduction section which claims a very slow behavioral change during crowding and a very rapid change during isolation.

10) How is the olfactory preference measured?

11) All immunostainings in Figure 8 seem overexposed making it unclear why specific regions and cells were selected over others.

12) I am always a little uneasy when western bolt data and their controls (tubulin as internal reference) are not shown on the same gel.

Discussion

1) As already mentioned, throughout there is a not well established connection between elevated locomotion activity and swarming (e.g. "anti-swarming"). This point deserves discussion.

2) Subsection “NPF1a and NPF2 act as anti-swarming neuromodulators in locusts through receptor-mediated pathways that alter NO levels”: the pars intercerebralis is not the same as the central complex (which is a major neuropil structure linked to control of locomotion in insects).

3) The previously mentioned issue of time courses deserves in-depth discussion.

4) I am missing some further discussion to put the current results in the context of the current knowledge on NPFs and feeding behavior. On the one hand, the suggested role for NPF in locust is inconsistent with the reported effects of this family of peptides on feeding behavior in other insects i.e. high levels of the peptides promote feeding. In locusts, it is the gregarious phase which is notorious for intense feeding behavior. However here NPF levels are low in gregarious animals. On the other hand, in *Drosophila* larva, NPF was found to be highly expressed in the brain during the feeding stage but not in the older larval brain during the wandering stage i.e. it has a suppressing effect on locomotion as suggested here.

5) Very strong support for the current findings come from Lucas et al. 2010 who reported a phase-dependent difference related to the foraging gene and cGMP-dependent protein kinase (PKG) in the desert locust. These authors found higher levels of PKG activity (the product of the foraging gene) in brains of gregarious locusts. PKG is known to be part of the Nitric Oxide/cGMP Signaling pathway, and indeed the current paper reports high NOS and NO levels in gregarious locusts. Moreover Lucas et al. observed high PKG activity and FOR expression in the anterior midline of the brain, the pars intercerebralis, consistent with the expression of NOS and of the NPF peptides reported in the current work. This work should be discussed here.

Materials and methods

6) While gregarious locusts were reared 500-1000 per cage, the crowding effect was achieved by placing 22 locust (2 S, 20 G) in the same size cage. This is a little troubling.

7) As already noted, in most cases data is not presented independently for attraction and for locomotor activity (since the measure for attraction is incorporated in the logistic regression model).

My last comment is that the manuscript will benefit from careful English proof reading (I do not provide specific examples to avoid a situation in which only my few example will be corrected).

One (after last) suggestion is to change the title to better reflect the work and the major findings. The current title suggests to the reader that the effect of NPF is already known and that the novelty is only in the role of NO. A better title will actually be more similar to the Impact statement.

---

## [Author Response]

*Essential revisions:*

*Reviewer #1:*

*The methodology and controls are thorough and the results regarding the impact of NPF peptides on NOS and NO are clear. However, it is unclear from the behavior presented whether the decreased/increased movement observed is directly related to swarming behavior, or rather a general increase in mobility. Obviously, increased mobility is needed for swarming, but the two are separate phenotypes. For example, increased locomotion would also affect a number of other behaviors, from foraging to mating, even though the authors show that, for example, it doesn't relate to changes in olfactory attraction to odors.*

We appreciate these comments and accept the reviewer’s opinion about the exact function of NPF/NO pathway on locomotion but not directly swarming behavior. We have revised our description throughout the text to make a clear emphasis on the regulation of phase-related locomotor plasticity of the NPF/NO pathway. We also changed the title of manuscript as: “The neuropeptide F/nitric oxide pathway is essential for shaping locomotor plasticity underlying locust phase transition”.

*Second, the results, in my interpretation, are presented as though the NPF / NO pathways observed lead to the phase change, while they may rather be a byproduct of the transition itself. This issue could be alleviated by a deep dive into the literature, including discussion of previous publications by the authors (some of which are already included in the references). For example:*

[…]

*These aspects of the results should be discussed to better interpret the results in light of current knowledge.*

We really appreciate the reviewer’s helpful suggestions. We have improved the discussion by explaining our results in the context of previously reported knowledge according to the reviewer’s suggestions. We have reorganized the sections and included new contents that address each issue concerned by the reviewer.

Detailed information is shown as following:

1) For the first point concerned by the review, we have added a new section to discuss the regulatory role of NPF in phase-related locomotion and its potential involvement in the phase-related behavioral choice of food quality (subsection “The NPF/NO signaling pathway plays an essential role in phase-related locomotor plasticity in locusts). Besides, the relationship between NPF/NO pathway and other phase-related characteristics has been discussed in subsection “Specific effects of different neuromodulators are essential for orchestrating phase-related behavioral traits”.

2) For the second and the third points, we have included additional text to compare NO-regulated locomotion with established roles of NO signaling in behavioral regulation in orthopteran and other insect species (subsection “The NPF/NO signaling pathway plays an essential role in phase-related locomotor plasticity in locusts”). Meanwhile, we also discussed the possible interaction between NO and other modulators (like dopamine and PKA) in the modulation of behavioral phase transition (subsection “Specific effects of different neuromodulators are essential for orchestrating phase-related behavioral traits”).

3) For the last point, we have provided the detailed discussion about the modulation of NO/cGMP/PKG signaling in behavioral phase transition of the two locust species, L. migratoria and S. gregaria, subsection “The NPF/NO signaling pathway plays an essential role in phase-related locomotor plasticity in locusts”.

*Reviewer #2:*

*My main criticism of this paper is that the Discussion focuses too much on reiterating the results, rather than focusing more on a synthetic interpretation of the results. This is reflected in the model presented in Figure 11, which provides little insight into how the neuropeptides, receptors and NOS enzyme act in concert to regulate locomotion through nitric oxide signalling. The model as presented suggests that p-NOS and NOS are equally active, and ignores much of the dynamics of the NPFs and receptors described in the results. To this reviewer, the results are consistent with the hypothesis that only p-NOS is active, and that NPF2 increases NOS activity only by providing a greater level of NOS substrate for phosphorylation. Also, that NPF1a-regulated NOS phosphorylation cycles quickly whereas NOS expression is slower to respond to NPF2 regulation not only explains the speed of locomotory transition between phases, but perhaps also explains why regulation of NPYR expression is necessary in the G-Phase to increase locomotion.*

We appreciate the reviewer’s comments. We have greatly revised the discussion in ways that should meet the reviewer’s concerns. In addition, we have revised the model in Figure 11 (and we certainly agree that only p-NOS is active, or at least that it is much more active than un-phosphorylated NOS).

*It would also help the reader if the effects of neuropeptide and NO signalling described in this paper could be put in context with the other hormonal and transcriptional regulators described previously. Where does the regulatory system described in this paper fit more broadly within the overall regulation of phase transition?*

We thank the review’s kindly suggestions. We have included additional discussion material on the NPF/NO pathway and on previously reported neuromodulators during the locust phase transition.

*Reviewer #3:*

*This paper describes a very thorough investigation suggesting a potential role for NPF in locust density-dependent phase transition and in locust phase characteristics.*

*The paper is overall well written and the findings well described. I do have several comments on the study, the text and figures. The following (a mix of major and minor comments) is arranged by section:*

*Introduction*

*1) While this is not a focus of this work, the authors tend to use "swarming" and "collective motion" interchangeably, while these refer to different aspects of locust (and other animals) behavior. A related point is that the gregarious phase is not a characteristic of a swarming population, but is rather characterized by swarming behavior. This may seem as merely semantics, yet in a paper describing the control of phase-transformation, I believe it is important. The authors are referred to Ariel and Ayali, 2015, for a recent review addressing these points.*

We agree with the reviewer’s suggestions. A key sentence has been revised to read “Swarming occurs in a wide variety of animal taxa, including insects, fish, birds, and mammals. Individuals benefit from swarming in many aspects, including food searching, territory selection, and defense”. We also revised the description as “the latter of which is characterized by swarming behavior (Ariel and Ayali, 2015)”.

*2) I am not sure that the sentence "Switching between key types of neuromodulators[…]" reflects the major points presented in the cited references, nor in the current text; the point being that neuromodulators are able of choosing among a behavioral repertoire and thus mediate behavioral plasticity. Similarly, neuropeptides induce behavioral plasticity rather than "modulate different types of behavioral plasticity".*

We have revised the sentence to read “Biochemical changes of neuromodulators, such as monoamines, neuropeptides, and neurohormones, are able to induce behavioral variation thus mediating behavioral plasticity”. Also, the description has been revised as the reviewer’s suggestion.

*3) The time course of a locust phase change is a repeated important aspect when reviewing the different presented results. It may deserve some further details and references (e.g. is gregarization indeed "initiates at least 32 h after crowding", or is it completed?)*

We agree with the reviewer’s suggestion. We have included additional descriptions as “Behavioral solitarization occurs faster than behavioral gregarization in the migratory locust. The attraction index and locomotor activity of locusts continuously decrease within 16 h after isolation. In contrast, these behaviors do not increase until 32 h after crowding, but are far below the level of gregarious controls even crowding for 64 h (Guo et al., 2011)”.

*4) While in the Abstract the authors state that the "neuropeptide F (NPF)-nitric oxide (NO) pathway plays a critical role in locomotor plasticity", in the Introduction they claim that the neuropeptides are crucial neuromodulators in "behavioral plasticity underlying swarming". As mentioned in my point 1 above, the connection between enhanced locomotion and swarming is not trivial.*

We have revised the sentence as “NPF1a and NPF2, act as crucial neural modulators in phase-related locomotor plasticity of the migratory locust”.

*Results*

*5) This is somewhat related to the issue of time courses mentioned earlier. Figure 1—figure supplement 1 shows 11 peptides' mRNA levels that do not change during phase transformation. According to this figure they also seem not to differ between the extreme phases. Yet, they were reported (and therefore presented here) as differentially expressed in gregarious and solitary locusts. Are these data contradicting? Should be explained/discussed. Furthermore, should one expect the levels of mRNA of peptides 0 hr after crowding to be similar to those of 32 hr after isolation and vice versa? This is not the case in Figure 1 or Figure 1—figure supplement 1 (nor in Figure 3, but compare to Figure 5).*

We have updated Figure 1—figure supplement 1 to present the phase-dependent differences in the expression levels of these neuropeptide genes. Actually, nine of the 11 genes (shown in Figure 1—figure supplement 1) have significantly different transcript levels between G-phase and S-phase locusts. Although the transcript levels of the other 2 genes do not differ significantly between two phases, the tendency of their expressions is consistent with our previous finding (Hou et al., 2015). In this study, we only compare the gene expressions between the typical G-phase (0 h after isolation) and typical S-phase (0 h after crowding) and the time-course expression changes of these neuropeptide genes during isolation or crowding. We did not compare the mRNA levels of peptide between 0 h after crowding and 32 h after isolation.

*Last, based on Figure 1, an effect of crowding is seen more often than an effect of isolation and usually within 1-4 hr. How does this fact correspond with the claim that in L. migratoria crowding is very quick while isolation is slow? All this points need clarification/discussion.*

We appreciate the reviewer’s suggestions. We have included additional discussion on the time-course changes of two NPFs and their receptors and NO levels, paragraph four subsection“NPF1a and NPF2, their receptors and NOS act in concert to regulate phase-related locomotion through NO signalling”.

*6) The authors present two measures of behavioral phase (e.g. Figure 2): the one is Pgreg and the other distance/duration of movement. The reader may tend to take these as a measure of activity (the latter) and a measure of attraction to peers (the former). However, Pgreg is a result of a binary logistic regression model which includes or is heavily based on the distance/duration of movement! Are locusts injected with NPF only less active or also less attracted to peers? This point brings me back to the issue of an effect on swarming behavior vs. a mere effect on locomotion, which are not the same. Another measure appears in the supplementary figures – Attraction index. There is no definition or explanation for how this measure is calculated.*

We have revised the description in the figure legend for Figure 2 as “Locust behaviors are measured as Pgreg, which is an combined assessment of movement and inter-insect attraction (indicated as attraction index, see Figure 2—figure supplement 4)”.

Our results show that two NPF peptides strongly regulate phase-related locomotor activity, yet do not affect attraction index of tested insects. We agree that the locomotion is not same as swarming behavior. Thus, we have revised the description throughout the text to appropriately present our finding by emphasizing the regulatory roles of NPF-NO pathway in locomotor plasticity during phase transition. Revisions are in the Introduction, Results, Discussion as well as the title.

*7) The most left columns (dsGFP) in Figure 2 seem to be similar. Why didn't crowding affect the control locusts (Figure 2)?*

Our previous study has shown that behavioral gregarization begins at least 32 h after crowding (Guo et al., 2011). In this study, the locusts were crowded for 16 h, thus we did not observe significant behavioral change in the dsGFP treated locusts.

*8) Not as described in subsection “Two NPF receptors, NPFR and NPYR, are essential for changes in locomotor activity related to phase transition”, according to Figure 3, the pattern of both NPFR and NPYR during isolation is very similar (increase) and somewhat similar during crowding (no change vs decrease)*

We have revised the sentence to read “In contrast, the transcript level of NPYR responded to both isolation and crowding, with obvious increase during isolation and significant reduction during crowding”.

*9) Coming back yet again to the important question of time courses, the authors state in subsection “NO signaling is a vital modulator of locomotor plasticity in the G/S phase transition” that "The mRNA and protein levels of NOS.… showed strong correlation with the time course of G/S phase transition". This statement is not consistent with the Introduction section which claims a very slow behavioral change during crowding and a very rapid change during isolation.*

We have revised the description as “The mRNA and protein levels of NOS were considerably higher in G-phase than in S-phase locust brains (Figure 5), and significantly changed during the time course of G/S phase transition (Figure 5)”. In addition, we have provided detailed discussion on the time-course changes of two NPFs and their receptors, NOS phosphorylation and NO levels.

*10) How is the olfactory preference measured?*

It should be “attraction index” instead of “olfactory preference”. Detailed method for the measurement of attraction index is shown in corresponding figure legend and the Materials and methods section.

*11) All immunostainings in Figure 8 seem overexposed making it unclear why specific regions and cells were selected over others.*

The immunostaining for each signal was exposed under the same condition with the negative control. Since NOS is most intensively stained in the pars intercerebralis, we thus only labeled several cells strongly stained by NOS and NPF peptide to present their co-localization in the pars intercerebralis. The overlap of NPF1a and NOS is also detected in the Kenyon cells anterior to the calyces of mushroom bodies in each brain hemisphere (please see Figure 8—figure supplement 1).

*12) I am always a little uneasy when western bolt data and their controls (tubulin as internal reference) are not shown on the same gel.*

We accept the query raised by the review. The western blot data for target protein and corresponding controls are from the same gel. Since the tubulin shows much higher expression level than the target protein in vivo, the membrane transferred with proteins was divided to two parts, and then incubated with different antibodies and visualized separately. Therefore, we did not show the data in the same figure. We have included detailed description on the methods of western blot.

*Discussion*

*1) As already mentioned, throughout there is a not well established connection between elevated locomotion activity and swarming (e.g. "anti-swarming"). This point deserves discussion.*

We have revised our description throughout the text to make a clear emphasis on phase-related locomotor plasticity instead of swarming behavior.

*2) Subsection “NPF1a and NPF2 act as anti-swarming neuromodulators in locusts through receptor-mediated pathways that alter NO levels”: the pars intercerebralis is not the same as the central complex (which is a major neuropil structure linked to control of locomotion in insects).*

We agree with the reviewer’s comment. We have revised our description and added new reference to support our discussion, subsection “NPF1a and NPF2, their receptors and NOS act in concert to regulate phase-related locomotion through NO signalling”.

*3) The previously mentioned issue of time courses deserves in-depth discussion.*

We have provided additional discussion on the time course changes of NPF/NO pathway during locust phase transition in paragraph four of subsection “NPF1a and NPF2, their receptors and NOS act in concert to regulate phase-related locomotion through NO signalling”.

*4) I am missing some further discussion to put the current results in the context of the current knowledge on NPFs and feeding behavior. On the one hand, the suggested role for NPF in locust is inconsistent with the reported effects of this family of peptides on feeding behavior in other insects i.e. high levels of the peptides promote feeding. In locusts, it is the gregarious phase which is notorious for intense feeding behavior. However here NPF levels are low in gregarious animals. On the other hand, in Drosophila larva, NPF was found to be highly expressed in the brain during the feeding stage but not in the older larval brain during the wandering stage i.e. it has a suppressing effect on locomotion as suggested here.*

We accept the query raised by the reviewer. We have put in additional discussion on this issue into subsection “The NPF/NO signaling pathway plays an essential role in phase-related locomotor plasticity in locusts”.

*5) Very strong support for the current findings come from Lucas et al. 2010 who reported a phase-dependent difference related to the foraging gene and cGMP-dependent protein kinase (PKG) in the desert locust. These authors found higher levels of PKG activity (the product of the foraging gene) in brains of gregarious locusts. PKG is known to be part of the Nitric Oxide/cGMP Signaling pathway, and indeed the current paper reports high NOS and NO levels in gregarious locusts. Moreover Lucas et al. observed high PKG activity and FOR expression in the anterior midline of the brain, the pars intercerebralis, consistent with the expression of NOS and of the NPF peptides reported in the current work. This work should be discussed here.*

We appreciate this suggestion. We have make a detailed comparison between the significance of NO in the migratory locust and the observations concerning PKG activity in the desert locust (Locas et al., 2010; Rogers et al. 2004).

*Materials and methods*

*6) While gregarious locusts were reared 500-1000 per cage, the crowding effect was achieved by placing 22 locust (2 S, 20 G) in the same size cage. This is a little troubling.*

The crowding treatment is conducted following our previous study (Guo et al., 2011). For gregarization, two fourth-instar S-phase nymphs were reared in small cage (10 cm × 10 cm × 10 cm) containing 20 G-phase locusts of the same developmental stage.

*7) As already noted, in most cases data is not presented independently for attraction and for locomotor activity (since the measure for attraction is incorporated in the logistic regression model)*

We have presented the measurements of locomotor activity and attraction index independently. The attraction index did not show significant change therefore the data is mainly shown in the supplemental figures. In this revision, we have provided detailed information on the measurement of attraction index in both Methods section and the figure legend.

*My last comment is that the manuscript will benefit from careful English proof reading (I do not provide specific examples to avoid a situation in which only my few example will be corrected).*

We have revised our writing with the help of an American colleague.

*One (after last) suggestion is to change the title to better reflect the work and the major findings. The current title suggests to the reader that the effect of NPF is already known and that the novelty is only in the role of NO. A better title will actually be more similar to the Impact statement.*

We appreciate the reviewer’s suggestion. We have changed the title to read “The neuropeptide F/nitric oxide pathway is essential for shaping locomotor plasticity underlying locust phase transition”.